# Thermodynamic limit and boundary energy of the spin-1 Heisenberg chain with non-diagonal boundary fields

Zhihan Zheng[1,2], Pei Sun[1,2], Xiaotian Xu[1,2*], Tao Yang[1,2,3,4], Junpeng Cao[4,5,6,7], Wen-Li Yang[1,2,3,4]

**1** Institute of Modern Physics, Northwest University, Xi'an 710127, China
**2** Shaanxi Key Laboratory for Theoretical Physics Frontiers, Xi'an 710127, China
**3** School of Physics, Northwest University, Xi'an 710127, China
**4** Peng Huanwu Center for Fundamental Theory, Xi'an 710127, China
**5** Beijing National Laboratory for Condensed Matter Physics, Institute of Physics, Chinese Academy of Sciences, Beijing 100190, China
**6** Songshan Lake Materials Laboratory, Dongguan, Guangdong 523808, China
**7** School of Physical Sciences, University of Chinese Academy of Sciences, Beijing 100049, China
*xtxu@nwu.edu.cn

## 1 Abstract

**The thermodynamic limit and boundary energy of the isotropic spin-1 Heisenberg chain with non-diagonal boundary fields are studied. The finite size scaling properties of the inhomogeneous term in the $T - Q$ relation at the ground state are calculated by the density matrix renormalization group. Based on our findings, the boundary energy of the system in the thermodynamic limit can be obtained from Bethe ansatz equations of a related model with parallel boundary fields. These results can be generalized to the $SU(2)$ symmetric high spin Heisenberg model directly.**

---

## Contents

---

## 1 Introduction

The study of quantum integrable models is an interesting subject in the fields of cold atoms, quantum field theory, condensed matter physics and statistic mechanics [1–5]. The

23 spin-1/2 Heisenberg model can effectively quantify the spin-exchanging interaction and
24 plays an important role in the quantum magnetism and many-body theory. By using
25 the Bethe ansatz method, the one-dimensional (1D) spin-1/2 Heisenberg model can be
26 solved exactly [6]. The typical spin-exchanging couplings in the 1D spin-1 system are
27 characterized by the bilinear biquadratic model, where the Hamiltonian reads

$$H = \sum_{k=1}^{N} \left[ J_1 \vec{S}_k \cdot \vec{S}_{k+1} + J_2 (\vec{S}_k \cdot \vec{S}_{k+1})^2 \right]. \tag{1}$$

28 Here $\vec{S}_k(S_k^x, S_k^y, S_k^z)$ is the spin-1 operator at site $k$, $N$ is the number of sites, and the
29 periodic boundary condition gives $\vec{S}_{N+1} = \vec{S}_1$. If $J_2/J_1 = 1$, the system (1) has the
30 $SU(3)$ symmetry and is integrable. If $J_2/J_1 = -1$, the $SU(2)$ symmetry exists, and
31 the system is known as the Zamalodchikov-Fateev (ZF) model [7]. The Bethe ansatz
32 solution and thermodynamic properties of the ZF model are studied by Takhtajan [8] and
33 Babujian [9,10]. If $J_2 = 0$, the system is no longer integrable. Starting from the nonlinear
34 sigma model, Haldane conjectures that the excitation of the system has a gap [11,12]. If
35 $J_2/J_1 = 1/3$, the Hamiltonian (1) degenerates into a projector operator that is in fact
36 the projection onto the sum of the spin-0 and spin-1 subspaces (up to a constant) and
37 the ground state is the famous valence bond solid state [13,14]. If $J_1 = 0$, by using the
38 Temperley-Lieb algebra, the system can be mapped into the XXZ spin chain and is also
39 integrable [15–17].
40    Besides the periodic boundary condition, the integrable open one is also an interesting
41 subject, which means that the system has magnetic impurity or the boundary magnetic
42 fields [18,19]. In the past few decades, the exact results of high spin models with periodic
43 [7–10, 20–25] and parallel boundary fields [26–29] have been extensively studied. It is
44 emphasized that the integrable boundary reflection matrix can have non-diagonal elements,
45 which means that the boundary fields are unparallel. Then the $U(1)$ symmetry is broken
46 and it is very hard to study the exact solution of the system. It is known that the integrable
47 systems without $U(1)$ symmetry have many applications in the open string theory and
48 the stochastic process of nonequilibrium statistics. Therefore, many interesting works of
49 high spin models with non-diagonal boundary reflections have been done [30–35].
50    Many attentions have been paid for quantum integrable models without $U(1)$ sym-
51 metry during past decades [36–49]. Recently, a systematic method, i.e., the off-diagonal
52 Bethe ansatz (ODBA) is proposed to solve the models with or without $U(1)$ symmetry [50].
53 Eigenvalues and eigenstates of several typical integrable models are obtained, where eigen-
54 values are given in terms of some homogeneous/inhomogeneous $T - Q$ relation [50–53].
55 The next task is to derive the physical quantities in the thermodynamic limit, which is
56 very complicated because the related Bethe ansatz equations (BAEs) are inhomogeneous
57 and the traditional thermodynamic Bethe ansatz can not be employed. In order to over-
58 come this difficulty, an effective method is to study the finite size scaling effects of the
59 inhomogeneous term in the $T - Q$ relation. With the help of this idea, the thermody-
60 namic limit, surface energy and elementary excitations of spin-1/2 XXZ spin chain with
61 arbitrary boundary fields are studied [54]. The boundary energy of the $SU(3)$ symmet-
62 ric spin-1 chain with generic integrable open boundaries is also obtained [55]. However,
63 the corresponding thermodynamic properties of the $SU(2)$ symmetric spin-1 Heisenberg
64 model are still missing.
65    In this paper, we study the thermodynamic limit and boundary energy of the spin-1
66 isotropic Heisenberg spin chain with non-diagonal boundary reflections. The finite size
67 scaling analysis of the contribution of the inhomogeneous term in the $T - Q$ relation
68 (namely, the third term in (18) below) to the ground state energy is studied as follows.

We first introduce a very function $\Lambda_{hom}(u)$ which is given in terms of a reduced $T - Q$ relation[1] (see (27) and (28) below) [51–53] and the associated BAEs are homogeneous ones (see (29) below). For any finite $N$, $\Lambda_{hom}(u)$ is *actually* not an eigenvalue of the transfer matrix with generic off-diagonal boundary $K$-matrices. Since that the function is given by a homogeneous $T - Q$ relation, we can apply the conventional thermodynamic Bethe ansatz [2] to investigate its thermodynamic limit. Then, comparing with the result of its thermodynamic limit and that of the density matrix renormalization group (DMRG) numerical [56–58] studies, we conclude that $\Lambda_{hom}(u)$, in the limit $N \to \infty$, really gives the correct boundary energy. Moreover, we find that most Bethe roots of the reduced BAEs at the ground state in the thermodynamic limit form 2-strings, associated with certain boundary strings and the rearrangement of the Fermi sea. The different structures of Bethe roots in different regimes of model parameters are given explicitly. Based on them, we obtain the boundary energy induced by the boundary magnetic fields. We also check the analytic results by the numerical extrapolation, and find that the analytical results and the numerical ones coincide with each other very well. The results given in this paper can be generalized to the $SU(2)$ symmetric spin-$s$ Heisenberg model directly.

This paper is organized as follows. Section 2 serves as an introduction to the notations for the spin-1 Heisenberg model with non-diagonal boundary fields. The ODBA exact solution is also briefly reviewed. In Section 3, we focus on the contribution of the inhomogeneous term in the $T - Q$ relation to the ground state energy. In Section 4, by using the patterns of Bethe roots of the reduced BAEs, we study the boundary energy of the model in the thermodynamic limit. We summarize the results and give some discussions in Section 5.

## 2  Non-diagonal boundary Spin-1 Heisenberg model

The spin-1 Heisenberg model with non-diagonal boundary fields is related to the 19-vertex $R$-matrix

$$R_{12}(u) = \begin{pmatrix} c(u) & & & & & & & & \\ & b(u) & & e(u) & & & & & \\ & & d(u) & & g(u) & & f(u) & & \\ & e(u) & & b(u) & & & & & \\ & & g(u) & & a(u) & & g(u) & & \\ & & & & & b(u) & & e(u) & \\ & & f(u) & & g(u) & & d(u) & & \\ & & & & & e(u) & & b(u) & \\ & & & & & & & & c(u) \end{pmatrix}, \qquad (2)$$

where the non-vanishing elements are

$$a(u) = u(u + \eta) + 2\eta^2, \ b(u) = u(u + \eta), \ c(u) = (u + \eta)(u + 2\eta),$$
$$d(u) = u(u - \eta), \ e(u) = 2\eta(u + \eta), \ f(u) = 2\eta^2, \ g(u) = 2u\eta, \qquad (3)$$

$u$ is the spectral parameter, and $\eta$ is the crossing parameter. Here we are dealing with the isotropic model, and $\eta$ can be scaled out. Throughout this paper, we adopt the standard notations. For any matrix $A \in \mathrm{End}(\mathbb{V})$, $A_j$ is an embedding operator in the tensor space $\mathbb{V} \otimes \mathbb{V} \otimes \cdots$, which acts as $A$ on the $j$-th space and as identity on the other factor spaces.

---

[1]The function $\Lambda_{hom}(u)$ can be simulated by eigenvalue of the transfer matrix with parallel boundary fields of the strengthes: $p \to p/\sqrt{1 + \alpha_-^2}$; $q \to q/\sqrt{1 + \alpha_+^2}$.

For any matrix $B \in \text{End}(\mathbb{V} \otimes \mathbb{V})$, $B_{i,j}$ is an embedding operator in the tensor space, which acts as an identity on the factor spaces except for the $i$-th and $j$-th ones. The $R$-matrix $R_{12}(u)$ satisfies the quantum Yang-Baxter equation (QYBE) [59, 60]

$$R_{12}(u - v)R_{13}(u)R_{23}(v) = R_{23}(v)R_{13}(u)R_{12}(u - v). \tag{4}$$

Besides, the $R$-matrix (2) also enjoys the properties

$$\text{Initial condition}: \ R_{12}(0) = 2\eta^2 P_{12}, \tag{5}$$

$$\text{Fusion condition}\ : \ R_{12}(-\eta) = 6\eta^2\, \mathbf{P}_{12}^{(0)}, \tag{6}$$

where $P_{12}$ is the permutation operator and $\mathbf{P}_{12}^{(0)}$ is the projector in the total spin-0 channel. The most general off-diagonal boundary reflection on one side of the chain is quantified by the reflection matrix obtained in [61, 62]

$$K^-(u) = (2u + \eta) \begin{pmatrix} x_1(u) & y_4'(u) & y_6'(u) \\ y_4(u) & x_2(u) & y_5'(u) \\ y_6(u) & y_5(u) & x_3(u) \end{pmatrix}, \tag{7}$$

where the matrix elements are

$$
\begin{aligned}
x_1(u) &= (p_- + u + \tfrac{\eta}{2})\,(p_- + u - \tfrac{\eta}{2}) + \frac{\alpha_-^2}{2}\,\eta\,(u - \tfrac{\eta}{2}), \\
x_2(u) &= (p_- + u - \tfrac{\eta}{2})\,(p_- - u + \tfrac{\eta}{2}) + \alpha_-^2\,(u + \tfrac{\eta}{2})\,(u - \tfrac{\eta}{2}), \\
x_3(u) &= (p_- - u - \tfrac{\eta}{2})\,(p_- - u + \tfrac{\eta}{2}) + \frac{\alpha_-^2}{2}\,\eta\,(u - \tfrac{\eta}{2}), \\
y_4(u) &= \sqrt{2}\,\alpha_-\,e^{-i\phi_-}\,u\,(p_- + u - \tfrac{\eta}{2}), \quad y_4'(u) = \sqrt{2}\,\alpha_-\,e^{i\phi_-}\,u\,(p_- + u - \tfrac{\eta}{2}), \\
y_5(u) &= \sqrt{2}\,\alpha_-\,e^{-i\phi_-}\,u\,(p_- - u + \tfrac{\eta}{2}), \quad y_5'(u) = \sqrt{2}\,\alpha_-\,e^{i\phi_-}\,u\,(p_- - u + \tfrac{\eta}{2}), \\
y_6(u) &= \alpha_-^2\,e^{-2i\phi_-}\,u\,(u - \tfrac{\eta}{2}), \quad y_6'(u) = \alpha_-^2\,e^{2i\phi_-}\,u\,(u - \tfrac{\eta}{2}),
\end{aligned} \tag{8}
$$

$p_-$, $\alpha_-$ and $\phi_-$ are the boundary parameters which measure the strength and direction of the boundary field. The reflection matrix $K^-(u)$ satisfies the reflection equation (RE)

$$R_{12}(u - v)K_1^-(u)R_{21}(u + v)K_2^-(v) = K_2^-(v)R_{21}(u + v)K_1^-(u)R_{12}(u - v). \tag{9}$$

The most general off-diagonal boundary reflection at the other side is quantified by the dual reflection matrix

$$K^+(u) = K^-(-u - \eta)\Big|_{(p_-,\alpha_-,\phi_-)\to(p_+,-\alpha_+,\phi_+)}, \tag{10}$$

where $p_+$, $\alpha_+$ and $\phi_+$ are the boundary parameters characterizing the strength and direction of the corresponding boundary field. The dual reflection matrix $K^+(u)$ satisfies the dual RE

$$
\begin{aligned}
R_{12}(v - u)&K_1^+(u)R_{21}(-u - v - 2\eta)K_2^+(v) \\
&= K_2^+(v)R_{21}(-u - v - 2\eta)K_1^+(u)R_{12}(v - u).
\end{aligned} \tag{11}
$$

From the $R$-matrix (2), we construct the single row monodromy matrices $T_0(u)$ and $\hat{T}_0(u)$ as

$$
\begin{aligned}
T_0(u) &= R_{0N}(u - \theta_N)R_{0N-1}(u - \theta_{N-1})\cdots R_{01}(u - \theta_1), \\
\hat{T}_0(u) &= R_{10}(u + \theta_1)R_{20}(u + \theta_2)\cdots R_{N0}(u + \theta_N),
\end{aligned} \tag{12}
$$

117  where $\{\theta_k, k = 1, \cdots, N\}$ are the inhomogeneous parameters, and the subscript 0 means
118  the auxiliary space and $1, \cdots, N$ denote the quantum spaces. The single row monodromy
119  matrices $T_0(u)$ and $\hat{T}_0(u)$ are the $3 \times 3$ matrices in the auxiliary space $\mathbf{V}_0$ and their elements
120  act on the quantum space $\mathbf{V}^{\otimes N}$. The transfer matrix of the system reads

$$t(u) = tr_0\{K_0^+(u)T_0(u)K_0^-(u)\hat{T}_0(u)\}. \tag{13}$$

121  From the QYBE (4), RE (9) and dual RE (11), one can prove that the transfer matrices
122  with different spectral parameters commute with each other, i.e.,

$$[t(u), t(v)] = 0. \tag{14}$$

123  Therefore, $t(u)$ serves as the generating functional of all the conserved quantities, which
124  ensures the integrability of the system. The model Hamiltonian is generated from the
125  transfer matrix $t(u)$ as [19]

$$
\begin{aligned}
H &= \partial_u \{\ln[t(u)]\}\big|_{u=0,\{\theta_k=0\}} \\
&= \frac{1}{\eta} \sum_{k=1}^{N-1} \left[ \vec{S}_k \cdot \vec{S}_{k+1} - (\vec{S}_k \cdot \vec{S}_{k+1})^2 \right] \\
&\quad + \frac{1}{p_-^2 - \frac{1}{4}\left(1 + \alpha_-^2\right)\eta^2} \left[ 2p_- \left(\alpha_- \cos\phi_- S_1^x - \alpha_- \sin\phi_- S_1^y + S_1^z\right) - \eta(S_1^z)^2 \right. \\
&\quad\quad - \frac{1}{2}\alpha_-^2 \eta \left[ \cos(2\phi_-) \left[(S_1^x)^2 - (S_1^y)^2\right] - (S_1^z)^2 \right] - \alpha_- \eta \cos\phi_- [S_1^x S_1^z + S_1^z S_1^x] \\
&\quad\quad \left. + \frac{1}{2}\alpha_-^2 \eta \sin(2\phi_-) [S_1^x S_1^y + S_1^y S_1^x] + \alpha_- \eta \sin\phi_- [S_1^y S_1^z + S_1^z S_1^y] \right] \\
&\quad + \frac{1}{p_+^2 - \frac{1}{4}\left(1 + \alpha_+^2\right)\eta^2} \left[ 2p_+ \left(\alpha_+ \cos\phi_+ S_N^x - \alpha_+ \sin\phi_+ S_N^y - S_N^z\right) - \eta\left(S_N^z\right)^2 \right. \\
&\quad\quad - \frac{1}{2}\alpha_+^2 \eta \left[ \cos(2\phi_+) \left[(S_N^x)^2 - (S_N^y)^2\right] - (S_N^z)^2 \right] + \alpha_+ \eta \cos\phi_+ [S_N^x S_N^z + S_N^z S_N^x] \\
&\quad\quad \left. + \frac{1}{2}\alpha_+^2 \eta \sin(2\phi_+) \left[S_N^x S_N^y + S_N^y S_N^x\right] - \alpha_+ \eta \sin\phi_+ \left[S_N^y S_N^z + S_N^z S_N^y\right] \right] \\
&\quad + \frac{\eta}{p_+^2 - \frac{1}{4}\left(1 + \alpha_+^2\right)\eta^2} + \frac{\eta}{p_-^2 - \frac{1}{4}\left(1 + \alpha_-^2\right)\eta^2} + \frac{1}{\eta}\left(3N + \frac{8}{3}\right).
\end{aligned}
\tag{15}
$$

126      Now, we seek the exact solution of the system (15). Let $|\Psi\rangle$ be an arbitrary eigenstate
127  of $t(u)$ with the eigenvalue $\Lambda(u)$, i.e.,

$$t(u)|\Psi\rangle = \Lambda(u)|\Psi\rangle. \tag{16}$$

128  Using the ODBA method [50] and fusion hierarchy, in the homogeneous limit $\{\theta_k = 0\}$,
129  the eigenvalue $\Lambda(u)$ can be expressed as the inhomogeneous $T - Q$ relation,

$$
\begin{aligned}
\Lambda(u) &= -4u(u + \eta)\Lambda^{(\frac{1}{2},1)}(u + \frac{\eta}{2})\Lambda^{(\frac{1}{2},1)}(u - \frac{\eta}{2}) + 4u(u + \eta)\delta^{(1)}(u + \frac{\eta}{2}), \tag{17} \\
\Lambda^{(\frac{1}{2},1)}(u) &= a^{(1)}(u)\frac{Q(u - \eta)}{Q(u)} + d^{(1)}(u)\frac{Q(u + \eta)}{Q(u)} + cu(u + \eta)\frac{F^{(1)}(u)}{Q(u)}, \tag{18}
\end{aligned}
$$

where

$$
\begin{aligned}
a^{(1)}(u) &= d^{(1)}(-u-\eta) \\
&= -\frac{2u+2\eta}{2u+\eta}(\sqrt{1+\alpha_+^2}\,u+p_+)(\sqrt{1+\alpha_-^2}\,u-p_-)\left(u+\frac{3\eta}{2}\right)^{2N}, \quad (19)
\end{aligned}
$$

$$
F^{(1)}(u) = (u-\frac{\eta}{2})^{2N}(u+\frac{\eta}{2})^{2N}(u+\frac{3\eta}{2})^{2N}, \tag{20}
$$

$$
\delta^{(1)}(u) = a^{(1)}(u)\,d^{(1)}(u-\eta), \tag{21}
$$

$$
c = 2\left[\alpha_-\alpha_+\cos(\phi_+-\phi_-)-1+\sqrt{(1+\alpha_-^2)(1+\alpha_+^2)}\right], \tag{22}
$$

$$
Q(u) = \prod_{k=1}^{2N}(u-u_k)(u+u_k+\eta) = Q(-u-\eta), \tag{23}
$$

and the $2N$ parameters $\{u_k|k=1,\cdots,2N\}$ in $Q$-function (23) are the Bethe roots. The singularity of eigenvalue $\Lambda(u)$ requires that the Bethe roots should satisfy the BAEs

$$
a^{(1)}(u_k)Q(u_k-\eta)+d^{(1)}(u_k)Q(u_k+\eta)+c\,u_k(u_k+\eta)\,F^{(1)}(u_k)=0, \quad k=1,\cdots,2N. \tag{24}
$$

The eigenvalue of Hamiltonian (15) reads

$$
E = \sum_{k=1}^{2N}\frac{4\eta}{(u_k+\frac{3\eta}{2})(u_k-\frac{\eta}{2})}+\frac{1}{\eta}3N+\frac{1}{\eta}E_0, \tag{25}
$$

where $\{u_k\}$ should satisfy the BAEs (24) and

$$
E_0 = \frac{8}{3}+\frac{2\sqrt{1+\alpha_+^2}\,p_+\eta}{p_+^2-\frac{\eta^2}{4}(1+\alpha_+^2)}-\frac{2\sqrt{1+\alpha_-^2}\,p_-\eta}{p_-^2-\frac{\eta^2}{4}(1+\alpha_-^2)}. \tag{26}
$$

Some remarks are in order. If the non-diagonal boundary parameters are $\alpha_+ = \alpha_- = 0$, or $\alpha_+ = -\alpha_- \neq 0$ and $\phi_- = \phi_+$ (which corresponds to the parallel boundary fields case), the parameter $c$ in Eq.(22) becomes zero and the corresponding $T-Q$ relation (18) is naturally reduced to the conventional diagonal one [30] obtained by the algebraic Bethe Ansatz.[2] For the other case with unparallel boundary fields, the parameter $c$ does not vanish. Thus the corresponding $T-Q$ relation has to include a non-vanishing inhomogeneous term for any finite $N$.

## 3   Finite size scaling behavior

The present BAEs (24) are inhomogeneous, thus it is very hard to investigate the thermodynamic properties of the system by using the traditional thermodynamic Bethe ansatz. In order to overcome this difficulty, we first analyze the contribution of inhomogeneous term in the $T-Q$ relation (18).

Define the reduced $T-Q$ relation as

$$
\Lambda_{hom}(u) = -4u(u+\eta)\Lambda_{hom}^{(\frac{1}{2},1)}(u+\frac{\eta}{2})\Lambda_{hom}^{(\frac{1}{2},1)}(u-\frac{\eta}{2})+4u(u+\eta)\delta^{(1)}(u+\frac{\eta}{2}), \tag{27}
$$

$$
\Lambda_{hom}^{(\frac{1}{2},1)}(u) = a^{(1)}(u)\frac{Q(u-\eta)}{Q(u)}+d^{(1)}(u)\frac{Q(u+\eta)}{Q(u)}. \tag{28}
$$

---

[2]If the non-diagonal boundary parameters satisfy the condition $\alpha_+ = \alpha_- \neq 0$, $|\phi_- - \phi_+| = \pi$ (which corresponds to the antiparallel boundary fields case), the parameter $c$ in Eq.(22) also becomes zero and the corresponding $T-Q$ relation naturally degenerates into the conventional diagonal one.

148 It should be emphasized that although the non-diagonal boundary parameters $\{p_\pm, \alpha_\pm\}$
149 except $\phi_\pm$ are included in the above reduced $T - Q$ relation (28), the $\Lambda_{hom}(u)$ is not
150 the eigenvalue $\Lambda(u)$ for any finite $N$ but rather that of the transfer matrix with parallel
151 boundary fields of the same strength. In the limit $N \to \infty$ it will give, however, the correct
152 boundary energy (see the following parts of the paper). From the singularity analysis of
153 the reduced $T - Q$ relation (28), we obtain the following reduced BAEs

$$\frac{\frac{i}{2} - \mu_k}{\frac{i}{2} + \mu_k} \frac{pi - \mu_k}{pi + \mu_k} \frac{qi - \mu_k}{qi + \mu_k} \left(\frac{i - \mu_k}{i + \mu_k}\right)^{2N} = \prod_{l=1}^{M} \frac{i - (\mu_k - \mu_l)}{i + (\mu_k - \mu_l)} \frac{i - (\mu_k + \mu_l)}{i + (\mu_k + \mu_l)}, \quad k = 1, \cdots, M, \quad (29)$$

154 where $M = 1, \cdots, 2N$ and we have put $\eta = 1$, $\mu_k = -iu_k - \frac{i}{2}$, $p = \frac{p_+}{\sqrt{1+\alpha_+^2}} - \frac{1}{2}$ and
155 $q = -\frac{p_-}{\sqrt{1+\alpha_-^2}} - \frac{1}{2}$ for convenience. From the $\Lambda_{hom}(u)$ given by Eq.(27), we obtain the
156 reduced energy which is defined as

$$E_{hom} = \partial_u \{\ln \Lambda_{hom}(u)\} \big|_{u=0} = -\sum_{k=1}^{M} \frac{4}{\mu_k^2 + 1} + 3N + E_0. \quad (30)$$

157 Solving the reduced BAEs (29), we could obtain the values of reduced Bethe roots $\{\mu_k\}$.
158 Substituting the Bethe roots into Eq.(30), we obtain the values of $E_{hom}$.
159    Let us focus on the ground state. The reduced ground state energy can be calculated
160 by the reduced BAEs (29). It is well-known that the even $N$ and odd $N$ give the same
161 physical properties in the thermodynamic limit. Thus we set $N$ as even. At the ground
162 state, the number of Bethe roots in the reduced BAEs (29) is $M = N$. For simplicity,
163 we choose the boundary parameters as $p > 0$ and $q \neq 0, -1$. We should note that at the
164 points of $q = 0, -1$, the boundary field is divergent due to the present parameterization of
165 the Hamiltonian (15). The distribution of reduced Bethe roots at the ground state in the
166 thermodynamic limit is shown in Figure 1. We see that the Bethe roots can be divided
167 into six different regimes in the $p - q$ plane.
168    1) In the regime I, where $p \geq 1/2$, $q < -1$, $-1/2 \leq q < 0$ or $q \geq 1/2$, all the Bethe
169 roots form 2-strings, i.e., $\mu_k = \lambda_k \pm \frac{i}{2} + \mathcal{O}(e^{-\delta N})$, where $\lambda_k$ denotes the position of 2-string
170 in the real axis, $\delta$ is a small positive number and $\mathcal{O}(e^{-\delta N})$ means the finite size correction.
171    2) In the regime II, where $p < 1/2$, $q < -1$, $-1/2 \leq q < 0$ or $q \geq 1/2$, besides $N - 2$
172 2-strings, there are two boundary strings, i.e., $pi$ and $(p-1)i$. The boundary strings mean
173 the pure imaginary Bethe roots which are related with the boundary parameters $p$ and
174 $q$ [63].
175    3) In the regime III, where $p \geq 1/2$ and $0 < q < 1/2$, besides $N - 2$ 2-strings, there
176 are two boundary strings, $qi$ and $(q - 1)i$.
177    4) In the regime IV, where $0 < p < 1/2$ and $0 < q < 1/2$, besides $N - 4$ 2-strings,
178 there are four boundary strings, $pi$, $(p-1)i$, $qi$ and $(q-1)i$.
179    5) In the regime V, where $p \geq 1/2$ and $-1 < q < -1/2$, besides $N - 2$ 2-strings, only
180 the boundary string $qi$ survives and one real Bethe root $\lambda_0$ appears which is caused by
181 the rearrangement of Fermi sea.
182    6) In the regime VI, where $0 < p < 1/2$ and $-1 < q < -1/2$, besides $N - 4$ 2-strings,
183 there are three boundary strings $qi$, $(q - 1)i$, $pi$ and one real root $\lambda_0$.
184    Because the Bethe roots are different in the different regimes of boundary parameters,
185 we shall discuss them separately. In the regime I, where all the Bethe roots are the
186 2-strings. Substituting the 2-string solutions into the reduced BAEs (29), omitting the
187 exponentially minor corrections and taking the product of all the string solutions, we

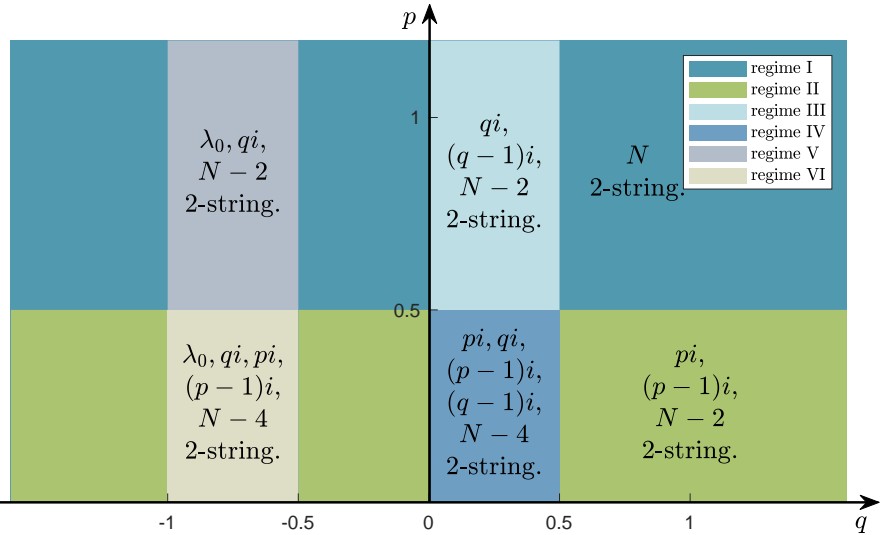

Figure 1: The distribution of reduced Bethe roots at the ground states with different boundary parameters $p$ and $q$.

readily obtain

$$
-\frac{i-\lambda_j}{i+\lambda_j}\frac{(p-\frac{1}{2})i-\lambda_j}{(p-\frac{1}{2})i+\lambda_j}\frac{(p+\frac{1}{2})i-\lambda_j}{(p+\frac{1}{2})i+\lambda_j}\frac{(q-\frac{1}{2})i-\lambda_j}{(q-\frac{1}{2})i+\lambda_j}\frac{(q+\frac{1}{2})i-\lambda_j}{(q+\frac{1}{2})i+\lambda_j}
$$
$$
\times\left(\frac{\frac{1}{2}i-\lambda_j}{\frac{1}{2}i+\lambda_j}\frac{\frac{3}{2}i-\lambda_j}{\frac{3}{2}i+\lambda_j}\right)^{2N}=\prod_{l=1}^{M_1}\left[\frac{i-(\lambda_j-\lambda_l)}{i+(\lambda_j-\lambda_l)}\right]^2\left[\frac{i-(\lambda_j+\lambda_l)}{i+(\lambda_j+\lambda_l)}\right]^2
$$
$$
\times\frac{2i-(\lambda_j-\lambda_l)}{2i+(\lambda_j-\lambda_l)}\frac{2i-(\lambda_j+\lambda_l)}{2i+(\lambda_j+\lambda_l)},\quad j=1,\cdots,M_1. \tag{31}
$$

Taking the logarithm of above Eq.(31), we obtain

$$
2\pi I_j=W(\lambda_j;M_1)+\theta_{2p-1}(\lambda_j)+\theta_{2p+1}(\lambda_j)+\theta_{2q-1}(\lambda_j)+\theta_{2q+1}(\lambda_j),\; j=1,\cdots,M_1, \tag{32}
$$

where

$$
W(\lambda_j;M_1)=\theta_2(\lambda_j)+2N\left[\theta_1(\lambda_j)+\theta_3(\lambda_j)\right]
$$
$$
-\sum_{l=1}^{M_1}\left[2\theta_2(\lambda_j-\lambda_l)+2\theta_2(\lambda_j+\lambda_l)+\theta_4(\lambda_j-\lambda_l)+\theta_4(\lambda_j+\lambda_l)\right], \tag{33}
$$

$I_j$ is the quantum number, $\theta_n(x)=2\arctan(2x/n)$ and $M_1=N/2$. The ground state is characterized by the set of quantum numbers

$$
\{I_j\}=\{1,2,\cdots,M_1\}. \tag{34}
$$

Solving the reduced BAEs (32) and substituting the values of Bethe roots into Eq.(30), we obtain the reduced ground state energy as

$$
E_{hom}=-2\sum_{j=1}^{M_1}\frac{1}{\lambda_j^2+\frac{1}{4}}+\frac{3}{\lambda_j^2+\frac{9}{4}}+3N+E_0\equiv G(\lambda_j;M_1). \tag{35}
$$

195  Now, we are ready to characterize the contribution of inhomogeneous term in the $T-Q$
196  relation (18) at the ground state by the quantity

$$E_{inh} = E_{hom} - E_g, \tag{36}$$

197  where $E_{hom}$ is the reduced ground state energy given by (35) and $E_g$ is the actual ground
198  state energy (25) of the Hamiltonian (15). The ground state energy $E_g$ can be obtained
199  by two methods. One is solving the inhomogeneous BAEs (24) directly and the other is
200  DMRG [56–58]. We have checked that the ground state energy $E$ obtained by these two
201  methods are the same.

202  In Figure 2(a), we give the values of $E_{inh}$ versus the system size $N$ in the regime I. The
203  red circles are the data calculated from Eq.(36) and the blue solid line is the fitted curve.
204  From the fitted curve, we find that $E_{inh}$ and $N$ satisfy the power law relation $E_{inh} = \gamma N^{\beta}$.
205  Due to the fact that $\beta < 0$, the value of $E_{inh}$ tends to zero when the system size $N$ tends
206  to infinity. Therefore, in the thermodynamic limit, the inhomogeneous term in the $T-Q$
207  relation (18) can be neglected at the ground state and $E_{hom} = E_g$. The inset shows the
208  distribution of Bethe roots with $N = 10$.

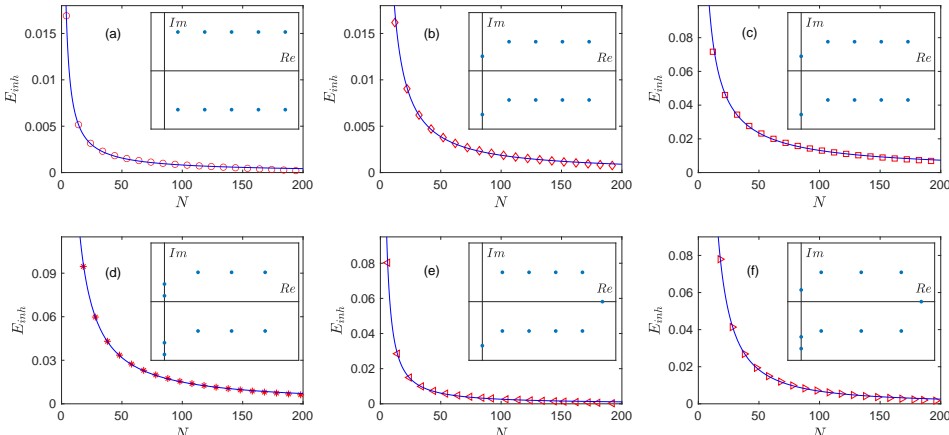

Figure 2: The values of $E_{inh}$ versus the system size $N$. The data can be fitted as $E_{inh} = \gamma N^{\beta}$. Due to the fact $\beta < 0$, when the size of system $N \to \infty$, the contribution of the inhomogeneous term tends to zero. Here (a) $p = 1.1370, q = -1.0821, \gamma = 0.06203$ and $\beta = -0.9407$ in regime I; (b) $p = 0.3263, q = -1.8931, \gamma = 0.2371$ and $\beta = -1.052$ in regime II; (c) $p = 0.2428, q = 2.3735, \gamma = 0.6236$ and $\beta = -0.8384$ in regime III; (d) $p = 0.4453, q = 0.3789, \gamma = 2.234$ and $\beta = -1.087$ in regime IV; (e) $p = 0.8410, q = -0.6990, \gamma = 0.715$ and $\beta = -1.219$ in regime V; (f) $p = 0.3971, q = -0.7985, \gamma = 4.912$ and $\beta = -1.429$ in regime VI. The insets show the distribution of Bethe roots with $N = 10$.

209  In the regime II, substituting the $N - 2$ 2-strings, two boundary strings $\mu_{M-1} = pi$
210  and $\mu_M = (p-1)i$ into the reduced BAEs (29) and taking the logarithm, we have

$$2\pi I_j = W(\lambda_j; M_2) + \theta_{2q-1}(\lambda_j) + \theta_{2q+1}(\lambda_j) - \theta_{1-2p}(\lambda_j) - \theta_{2p+1}(\lambda_j)$$
$$-\theta_{3+2p}(\lambda_j) - \theta_{5-2p}(\lambda_j) - 2\theta_{3-2p}(\lambda_j), \quad j = 1, 2, \cdots, M_2, \tag{37}$$

211  where $W(\lambda_j; M_2)$ is given by Eq.(33) with the replacing of $M_1$ by $M_2$, $M_2 = N/2 - 1$ and
212  the quantum numbers are

$$\{I_j\} = \{1, 2, \cdots, M_2\}. \tag{38}$$

213  The corresponding reduced ground state energy reads

$$E_{hom} = G(\lambda_j; M_2) + \frac{4}{p^2 - 1} + \frac{4}{(p-1)^2 - 1}, \tag{39}$$

214  where $G(\lambda_j; M_2)$ is given by Eq.(35) with the replacing of $M_1$ by $M_2$.

215  The procedure in the regime III is similar and reduced ground state energy is

$$E_{hom} = G(\lambda_j; M_2) + \frac{4}{q^2 - 1} + \frac{4}{(q-1)^2 - 1}. \tag{40}$$

216  In the regime IV, substituting the string solutions including four boundary strings into
217  Eq.(29) and taking the logarithm, we have

$$2\pi I_j = W(\lambda_j; M_3) - \theta_{1-2p}(\lambda_j) - \theta_{2p+1}(\lambda_j) - \theta_{3+2p}(\lambda_j) - \theta_{5-2p}(\lambda_j) - 2\theta_{3-2p}(\lambda_j)$$
$$-\theta_{1-2q}(\lambda_j) - \theta_{2q+1}(\lambda_j) - \theta_{3+2q}(\lambda_j) - \theta_{5-2q}(\lambda_j) - 2\theta_{3-2q}(\lambda_j), \quad j = 1, 2, \cdots, M_3, \tag{41}$$

218  where $M_3 = N/2 - 2$ and the quantum numbers are

$$\{I_j\} = \{1, 2, \cdots, M_3\}. \tag{42}$$

219  The reduced ground state energy is

$$E_{hom} = G(\lambda_j; M_3) + \frac{4}{p^2 - 1} + \frac{4}{(p-1)^2 - 1} + \frac{4}{q^2 - 1} + \frac{4}{(q-1)^2 - 1}. \tag{43}$$

220  In the regime V, the logarithm form of the BAEs are

$$2\pi I_j = W(\lambda_j; M_4) + \theta_{2p-1}(\lambda_j) + \theta_{2p+1}(\lambda_j) - \theta_{3+2q}(\lambda_j) - \theta_{3-2q}(\lambda_j) - 2\theta_{1-2q}(\lambda_j)$$
$$-\theta_1(\lambda_j - \lambda_0) - \theta_1(\lambda_j + \lambda_0) - \theta_3(\lambda_j - \lambda_0) - \theta_3(\lambda_j + \lambda_0), \quad j = 1, 2, \cdots, M_4, \tag{44}$$

221  where $M_4 = N/2 - 1$ and the quantum numbers are $\{I_j\} = \{1, 2, \cdots, M_4\}$. We shall
222  note that the quantum number corresponding to the real Bethe root $\lambda_0$ is 0. The reduced
223  ground state energy reads

$$E_{hom} = G(\lambda_j; M_4) + \frac{4}{q^2 - 1} - \frac{4}{\lambda_0^2 + 1}. \tag{45}$$

224  Similarly, the reduced ground state energy in the regime VI is

$$E_{hom} = G(\lambda_j; M_5) + \frac{4}{p^2 - 1} + \frac{4}{(p-1)^2 - 1} + \frac{4}{q^2 - 1} - \frac{4}{\lambda_0^2 + 1}, \tag{46}$$

225  where $M_5 = N/2 - 2$.

226  Substituting the reduced ground state energies in different regimes into Eq.(36), we
227  obtain the values of $E_{inh}$, which are shown in Figures 2(b)-(f). According to the finite
228  size scaling analysis, we see that the inhomogeneous term indeed can be neglected at the
229  ground state in the thermodynamic limit. Due to the existence of inhomogeneous term in
230  BAEs.(24), it is hard to analytically calculate the finite size correction for the present off-
231  diagonal boundary reflections along the lines given in references [64–66]. We shall note that
232  the diagonal case is tractable along the lines of A. Klümper et al. [65] and J. Suzuki [66].
233  The $\mathcal{O}(N^1)$ bulk term and the $\mathcal{O}(N^0)$ boundary term for the ground state energy do not
234  depend on the orientations of the boundary fields. The true finite size correction terms
235  are probably of order $\mathcal{O}(N^{-1})$ and are out of reach for the inhomogeneous/off-diagonal
236  case. Due to higher order correction terms, the effective exponents $\beta$ determined in the
237  paper differ from $-1$.

## 4 Boundary energy

In this section, we study the physical effects induced by the boundary magnetic fields and compute the boundary energy in the thermodynamic limit [18, 35, 67–69]. As mentioned above, we can calculate the boundary energy based on the string hypothesis of the reduced BAEs (29), then the numerical analysis allows us to obtain the boundary energy induced by the boundary fields.

The values of Bethe roots at the ground state are determined by the quantum numbers $\{I_j\}$. Thus we define the counting function as $Z(\lambda_j) = \frac{I_j}{2N}$. In the thermodynamic limit, the Bethe roots can take the continuous values and we have $Z(\lambda_j) \to Z(u)$. Taking the derivative of $Z(u)$ with respect to $u$, we obtain

$$\frac{dZ(u)}{du} = \rho(u) + \rho^h(u), \tag{47}$$

where $\rho(u)$ is the density of Bethe roots and $\rho^h(u)$ means the density of holes in the real axis. Again, the distribution of Bethe roots in different regimes are different. We should consider them separately. In regime I, from the BAEs (32) with the constraint $N \to \infty$ and using Eq.(47), we obtain the density of states as

$$
\begin{aligned}
\rho(u) &= \frac{dZ(u)}{du} - \frac{1}{2N}[\rho^h(u) + \delta(u)] \\
&= a_1(u) + a_3(u) + \frac{1}{2N}\left[a_2(u) + a_{2p-1}(u) + a_{2p+1}(u) + a_{2q-1}(u) + a_{2q+1}(u)\right] \\
&\quad - \frac{1}{2N}[\rho^h(u) + \delta(u)] - \int_{-\infty}^{\infty} \left[2a_2(u-v) + a_4(u+v)\right]\rho(v)dv,
\end{aligned}
\tag{48}
$$

where

$$a_n(u) = \frac{1}{2\pi}\frac{n}{u^2 + \frac{n^2}{4}},$$

$$\rho^h(u) = \frac{1}{2N}\left[\delta\left(u - \lambda_1^h\right) + \delta\left(u + \lambda_1^h\right) + \delta\left(u - \lambda_2^h\right) + \delta\left(u + \lambda_2^h\right)\right]. \tag{49}$$

We should note that the presence of delta-function in Eq.(48) is due to that $\lambda_j = 0$ is the solution of BAEs (32), which should be excluded because it makes the wavefunction vanish identically [70]. Note that two holes $\lambda_1^h$ and $\lambda_2^h$ are introduced to ensure the magnetization satisfying

$$\frac{M}{N} = 2\int_{-\infty}^{\infty} \rho(u)du = 1. \tag{50}$$

Thus the holes are located at the infinities in the real axis.

With the help of Fourier transformation

$$\tilde{F}(\omega) = \int_{-\infty}^{\infty} e^{i\omega u}F(u)du, \qquad F(u) = \frac{1}{2\pi}\int_{-\infty}^{\infty} e^{-i\omega u}\tilde{F}(\omega)d\omega, \tag{51}$$

from Eq.(48), we obtain

$$\tilde{\rho}(\omega) = \tilde{\rho}_g(\omega) + \tilde{\rho}_0(\omega) + \tilde{\rho}_1(\omega) + \tilde{\rho}_2(\omega), \tag{52}$$

where

$$
\tilde{a}_n(\omega) = e^{-\frac{n|\omega|}{2}}, \quad \tilde{\rho}_g(\omega) = \frac{\tilde{a}_1(\omega) + \tilde{a}_3(\omega)}{1 + 2\tilde{a}_2(\omega) + \tilde{a}_4(\omega)}, \quad \tilde{\rho}_0(\omega) = \frac{1}{2N} \frac{\tilde{a}_2(\omega) - 1}{1 + 2\tilde{a}_2(\omega) + \tilde{a}_4(\omega)},
$$

$$
\tilde{\rho}_1(\omega) = \begin{cases} \dfrac{1}{2N} \dfrac{\tilde{a}_{2p+1}(\omega) - \tilde{a}_{1-2p}(\omega)}{1 + 2\tilde{a}_2(\omega) + \tilde{a}_4(\omega)}, & 0 < p < \dfrac{1}{2}, \\[3mm] \dfrac{1}{2N} \dfrac{\tilde{a}_{2p-1}(\omega) + \tilde{a}_{2p+1}(\omega)}{1 + 2\tilde{a}_2(\omega) + \tilde{a}_4(\omega)}, & p > \dfrac{1}{2}, \end{cases}
$$

$$
\tilde{\rho}_2(\omega) = \begin{cases} -\dfrac{1}{2N} \dfrac{\tilde{a}_{1-2q}(\omega) + \tilde{a}_{-2q-1}(\omega)}{1 + 2\tilde{a}_2(\omega) + \tilde{a}_4(\omega)}, & q < -\dfrac{1}{2}, \\[3mm] \dfrac{1}{2N} \dfrac{\tilde{a}_{2q+1}(\omega) - \tilde{a}_{1-2q}(\omega)}{1 + 2\tilde{a}_2(\omega) + \tilde{a}_4(\omega)}, & -\dfrac{1}{2} < q < \dfrac{1}{2}, \\[3mm] \dfrac{1}{2N} \dfrac{\tilde{a}_{2q-1}(\omega) + \tilde{a}_{2q+1}(\omega)}{1 + 2\tilde{a}_2(\omega) + \tilde{a}_4(\omega)}, & q > \dfrac{1}{2}. \end{cases} \tag{53}
$$

Then the ground state energy (35) can be expressed as

$$
E_g = -2N \int_{-\infty}^{\infty} [\tilde{a}_1(\omega) + \tilde{a}_3(\omega)] \tilde{\rho}(\omega) d\omega + 3N + E_0 = Ne_g + e_s, \tag{54}
$$

where $e_g$ is the ground state energy density which is the same as that for the periodic boundary condition [9],

$$
e_g = -2 \int_{-\infty}^{\infty} \frac{[\tilde{a}_1(\omega) + \tilde{a}_3(\omega)]^2}{1 + 2\tilde{a}_2(\omega) + \tilde{a}_4(\omega)} d\omega + 3 = -1, \tag{55}
$$

and $e_s$ is boundary energy

$$
e_s = 2\pi - 4 + E_0 + e_1 + e_2, \tag{56}
$$

$$
e_1 = \begin{cases} -\displaystyle\int_{-\infty}^{\infty} [\tilde{a}_1(\omega) + \tilde{a}_3(\omega)] \dfrac{\tilde{a}_{2p-1}(\omega) + \tilde{a}_{2p+1}(\omega)}{1 + 2\tilde{a}_2(\omega) + \tilde{a}_4(\omega)} d\omega, & p > \dfrac{1}{2}, \\[4mm] -\displaystyle\int_{-\infty}^{\infty} [\tilde{a}_1(\omega) + \tilde{a}_3(\omega)] \dfrac{\tilde{a}_{2p+1}(\omega) - \tilde{a}_{1-2p}(\omega)}{1 + 2\tilde{a}_2(\omega) + \tilde{a}_4(\omega)} d\omega, & 0 < p < \dfrac{1}{2}, \end{cases} \tag{57}
$$

$$
e_2 = \begin{cases} \displaystyle\int_{-\infty}^{\infty} [\tilde{a}_1(\omega) + \tilde{a}_3(\omega)] \dfrac{\tilde{a}_{-2q-1}(\omega) + \tilde{a}_{1-2q}(\omega)}{1 + 2\tilde{a}_2(\omega) + \tilde{a}_4(\omega)} d\omega, & q < -\dfrac{1}{2}, \\[4mm] -\displaystyle\int_{-\infty}^{\infty} [\tilde{a}_1(\omega) + \tilde{a}_3(\omega)] \dfrac{\tilde{a}_{2q+1}(\omega) - \tilde{a}_{1-2q}(\omega)}{1 + 2\tilde{a}_2(\omega) + \tilde{a}_4(\omega)} d\omega, & -\dfrac{1}{2} < q < \dfrac{1}{2}, \\[4mm] -\displaystyle\int_{-\infty}^{\infty} [\tilde{a}_1(\omega) + \tilde{a}_3(\omega)] \dfrac{\tilde{a}_{2q-1}(\omega) + \tilde{a}_{2q+1}(\omega)}{1 + 2\tilde{a}_2(\omega) + \tilde{a}_4(\omega)} d\omega, & q > \dfrac{1}{2}. \end{cases} \tag{58}
$$

Now, we consider the regime II. The boundary strings $pi$ and $(p-1)i$ can give rise to the rearrangement of Bethe roots in Fermi sea. From BAEs (37), the density of states $\rho_p(u)$ is obtained as

$$
\begin{aligned}
\rho_p(u) &= a_1(u) + a_3(u) - \int_{-\infty}^{\infty} [2a_2(u-v) + a_4(u-v)] \rho_p(v) dv \\
&\quad + \frac{1}{2N} [a_2(u) - a_{1-2p}(u) + a_{2p+1}(u) + a_{2q-1}(u) + a_{2q+1}(u) - \delta(u)] \\
&\quad - \frac{1}{2N} [2a_{2p+1}(u) + 2a_{3-2p}(u) + a_{3+2p}(u) + a_{5-2p}(u)].
\end{aligned} \tag{59}
$$

In order to show that there exist the stable boundary bound states, we denote the deviation between $\rho_p(u)$ and $\rho(u)$ as $\delta\rho_p(u) = \rho_p(u) - \rho(u)$. From Eqs.(48) and (59), we obtain

$$
\begin{aligned}
\delta\rho_p(u) &= -\frac{1}{2N} \left[ 2a_{2p+1}(u) + 2a_{3-2p}(u) + a_{3+2p}(u) + a_{5-2p}(u) \right] \\
&\quad - \int_{-\infty}^{\infty} \left[ 2a_2(u-v) + a_4(u-v) \right] \delta\rho_p(v) dv.
\end{aligned}
\tag{60}
$$

Taking the Fourier transformation of Eq.(60), we have

$$
\delta\tilde{\rho}_p(\omega) = -\frac{1}{2N} \frac{2\tilde{a}_{2p+1}(\omega) + 2\tilde{a}_{3-2p}(\omega) + \tilde{a}_{3+2p}(\omega) + \tilde{a}_{5-2p}(\omega)}{1 + 2\tilde{a}_2(\omega) + \tilde{a}_4(\omega)}.
\tag{61}
$$

The energy deviation $\delta e_p$ induced by the density deviation $\delta\tilde{\rho}_p(\omega)$ can be expressed as

$$
\begin{aligned}
\delta e_p &= -2N \int_{-\infty}^{\infty} \left[ \tilde{a}_1(\omega) + \tilde{a}_3(\omega) \right] \delta\tilde{\rho}_p(\omega) d\omega + \frac{4}{p^2-1} + \frac{4}{(p-1)^2-1} \\
&= 2 \int_0^{\infty} \frac{e^{-(p+1)\omega}}{1+e^{-\omega}} dw + 2 \int_0^{\infty} \frac{e^{-(2-p)\omega}}{1+e^{-\omega}} d\omega + \frac{2}{p(p-1)} < 0.
\end{aligned}
\tag{62}
$$

Because of $\delta e_p < 0$, the boundary strings are stable. Then we conclude that in this regime, the ground state energy of the system is $E_g = Ne_g + e_s + \delta e_p$. The total spin along the $z$-direction is $S_z = -\int_{-\infty}^{\infty} \delta\rho_p(u) = 3/4$.

Next, we consider the regime III where boundary strings are $qi$ and $(q-1)i$. Similarly, the energy deviation $\delta e_q$ between this case and that without boundary strings is

$$
\begin{aligned}
\delta e_q &= -2N \int_{-\infty}^{\infty} \left[ \tilde{a}_1(\omega) + \tilde{a}_3(\omega) \right] \delta\tilde{\rho}_q(\omega) d\omega + \frac{4}{p^2-1} + \frac{4}{(p-1)^2-1} \\
&= 2 \int_0^{\infty} \frac{e^{-(q+1)\omega}}{1+e^{-\omega}} dw + 2 \int_0^{\infty} \frac{e^{-(2-q)\omega}}{1+e^{-\omega}} d\omega + \frac{2}{q(q-1)} < 0.
\end{aligned}
\tag{63}
$$

Due to the fact $\delta e_q < 0$, we know that the ground state energy is $E_g = Ne_g + e_s + \delta e_q$ and the total spin along the $z$-direction is $S_z = 3/4$.

In the regime IV, we combine the results (62) and (63), and conclude that the ground state energy with boundary strings $pi$, $(p-1)i$, $qi$ and $(q-1)i$ equals to $E_g = Ne_g + e_s + \delta e_p + \delta e_q$.

Then, we consider the regime V where besides the $N-2$ 2-string, there also exist one real Bethe root $\lambda_0$ and a single boundary string $qi$. Taking the thermodynamic limit of BAEs (44), we obtain the density of states $\rho_{\lambda q}(u)$ as

$$
\begin{aligned}
\rho_{\lambda q}(u) &= a_1(u) + a_3(u) - \frac{1}{2N} \left[ a_1(u-\lambda_0) + a_1(u+\lambda_0) + a_3(u-\lambda_0) + a_3(u+\lambda_0) \right] \\
&\quad + \frac{1}{2N} \left[ a_2(u) + a_{2p-1}(u) + a_{2p+1}(u) - 2a_{1-2q}(u) - a_{3+2q}(u) - a_{3-2q}(u) - \delta(u) \right] \\
&\quad - \int_{-\infty}^{\infty} \left[ 2a_2(u-v) + a_4(u-v) \right] \rho_{\lambda q}(v) dv.
\end{aligned}
\tag{64}
$$

Denote the deviation between $\rho_{\lambda q}(u)$ and $\rho(u)$ as $\delta\rho_{\lambda q}(u) = \rho_{\lambda q}(u) - \rho(u)$. From Eqs.(48) and (64), the value of $\delta\rho_{\lambda q}(u)$ reads

$$
\begin{aligned}
\delta\rho_{\lambda q}(u) &= -\frac{1}{2N} \left[ a_1(u-\lambda_0) + a_1(u+\lambda_0) + a_3(u-\lambda_0) + a_3(u+\lambda_0) \right] \\
&\quad - \frac{1}{2N} \left[ a_{1-2q}(u) - a_{-1-2q}(u) + a_{3-2q}(u) + a_{3+2q}(u) \right] \\
&\quad - \int_{-\infty}^{\infty} \left[ 2a_2(u) + a_4(u) \right] \delta\rho_{\lambda q}(v) dv.
\end{aligned}
\tag{65}
$$

287   Taking the Fourier transformation of Eq.(65), we obtain

$$\delta\tilde{\rho}_{\lambda q}(\omega) = -\frac{1}{2N}\frac{\tilde{a}_{1-2q}(\omega) - \tilde{a}_{-1-2q}(\omega) + \tilde{a}_{3-2q}(\omega) + \tilde{a}_{3+2q}(\omega)}{1 + 2\tilde{a}_2(\omega) + \tilde{a}_4(\omega)} - \frac{1}{N}\frac{\cos(\omega\lambda_0)e^{-\frac{|\omega|}{2}}}{1 + e^{-|\omega|}}. \quad (66)$$

288   Then the deviation of energy $\delta e_{\lambda q}$ induced by $\delta\tilde{\rho}_{\lambda q}(\omega)$ is given by

$$
\begin{aligned}
\delta e_{\lambda q} &= -2N\int_{-\infty}^{\infty}\left[\tilde{a}_1(\omega) + \tilde{a}_3(\omega)\right]\delta\tilde{\rho}_{\lambda q}(\omega)d\omega + \frac{4}{q^2 - 1} - \frac{4}{\lambda_0^2 + 1} \\
&= 2\int_0^{\infty}\frac{e^{-(2+q)\omega}}{1 + e^{-\omega}}d\omega - 2\int_0^{\infty}\frac{e^{q\omega}}{1 + e^{-\omega}}d\omega - \frac{2}{1 + q} < 0. \quad (67)
\end{aligned}
$$

289   Due to $\delta e_{\lambda q} < 0$, the ground state energy in this regime is $E_g = Ne_g + e_s + \delta e_{\lambda q}$ and the
290   total spin along the $z$-direction is $S_z = 3/4$.
291       In the regime VI, there are $N - 4$ 2-string, one real Bethe root $\lambda_0$ and three boundary
292   strings $qi$, $pi$ and $(p-1)i$. Combining the results (62) and (67), we obtain the ground
      state energy as $E_g = Ne_g + e_s + \delta e_p + \delta e_{\lambda q}$.

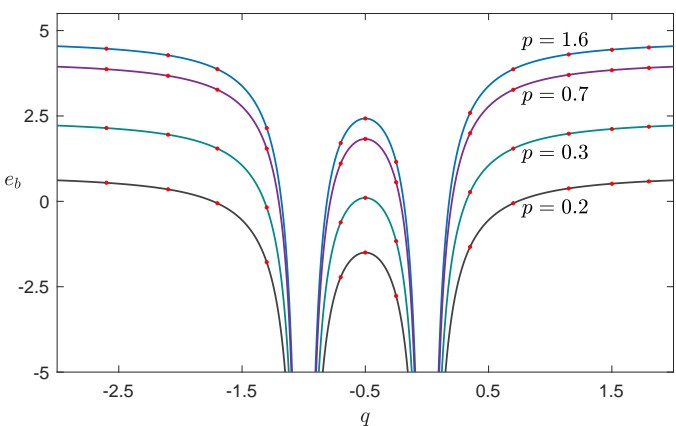

Figure 3: Boundary energies versus the boundary parameters $p$ and $q$. The coloured
curves are those calculated from the analytical expression (68) and the red points
are those obtained from the DMRG. The values of $q$ at the red points are $q = -2.6, -2.1, -1.7, -1.3, -0.7, -0.5, -0.25, 0.35, 0.7, 1.15, 1.5$ and $1.8$.

293
294       After tedious calculation, we find that the boundary energy $e_b$ for all the regimes in
295   Figure 1 can be expressed as

$$
e_b = \begin{cases} -\dfrac{2}{p} - \dfrac{2}{q} + 2\pi - 4 + E_0, & p > 0, \ q > 0 \text{ or } q < -1, \\[2ex] -\dfrac{2}{p} - \dfrac{2}{q} + 2\pi\csc(q\pi) + 2\pi - 4 + E_0, & p > 0, \ -1 < q < 0. \end{cases} \quad (68)
$$

296   The boundary energies with different boundary parameters $p$ and $q$ calculated by the
297   analytical expression (68) are shown in Figure 3 as the coloured solid lines. Now we check
298   the correction of expression (68) by the numerical simulation with DMRG algorithm, and
299   the results are shown in Figure 3 as the red points. Specifically, for each red point that
300   is for the given boundary parameters $p$ and $q$, we first calculate the ground state energy
301   $E_g(N)$ of the model (15) with the system size $N = 10(n-1) + 4$ and $n = 1, 2, \cdots, 20$ by
302   using the DMRG method. Then we consider the physical quantity

$$e_b(N) = E_g(N) - Ne_g, \quad (69)$$

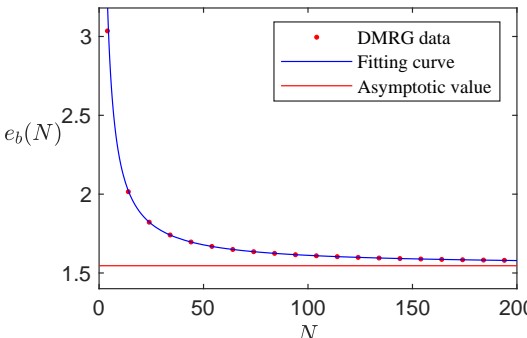

Figure 4: The values of $e_b(N)$ versus the system size $N$. The red points are the DMRG results with $N = 4, 14, 24, \cdots, 194$. The data can be fitted as $e_b(N) = aN^\beta + c$, where $a = 6.7308$, $\beta = -1.0046$ and $c = 1.5460$. Due to the fact $\beta < 0$, when the system size $N \to \infty$, the values of $e_b(N)$ tend to the asymptotic value $c$, which gives the boundary energy. Here the boundary parameters are chosen as $p = 0.3$ and $q = 0.7$.

where $e_g = -1$ is the ground state energy density of the system with periodic boundary conditions. Obviously, in the thermodynamic limit, the value of $e_b(N \to \infty)$ gives the boundary energy. In Figure 4, we show how to extrapolate the boundary energy, where the red points are the numerical values of $e_b(N)$, the blue solid line is the fitting curve, and the red solid line is the extrapolated boundary energy. From the fitting curve, we find that the $e_b(N)$ and $N$ satisfy the power law relation, i.e., $e_b(N) = aN^\beta + c$. Due to the fact that $\beta < 0$, the values of $e_b(N)$ tend to the asymptotic value $c$ when the system size $N$ tends to infinity. Therefore, in the thermodynamic limit, the asymptotic value $c$ determines the boundary energy. Repeating this process, we obtain the boundary energies with other values of boundary parameters. As shown in Figure 3, the analytical and numerical results agree with each other very well.

## 5    Conclusions

In this paper, we have studied the thermodynamic limit and boundary energy of the isotropic spin-1 Heisenberg chain with generic integrable non-diagonal boundary reflections. It is shown that the contribution of the inhomogeneous term in the associated $T - Q$ relation (18) (due to the unparallel boundary fields) at the ground state can be neglected when the system size $N$ tend to infinity. Then we calculate the analytical expression of boundary energy (68) in the thermodynamic limit based on the string hypothesis of the reduced BAEs (29).

## Acknowledgments

Financial support from the National Natural Science Foundation of China (Grant Nos. 12105221, 12175180, 12074410, 12047502, 11934015, 11975183, 11947301, 11775177, 11775178 and 11774397), the Strategic Priority Research Program of the Chinese Academy of Sciences (Grant No. XDB33000000), the Major Basic Research Program of Natural Science of Shaanxi Province (Grant Nos. 2021JCW-19, 2017KCT-12 and 2017ZDJC-32), the Scientific Research Program Funded by Shaanxi Provincial Education Department (Grant

No. 21JK0946), Beijing National Laboratory for Condensed Matter Physics (Grant No. 202162100001), and the Double First-Class University Construction Project of Northwest University is gratefully acknowledged. One of the authors, Zhihan Zheng would like to thank Dr. Yangyang Chen, Dr. Fakai Wen and Dr. Yi Qiao for their helpful discussions.

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
