# Peer review of "Thermodynamic limit and boundary energy of the spin-1 Heisenberg chain with non-diagonal boundary fields"

_SciPost Physics_

## Round 1 · Referee Report · Anonymous (Referee 1) · 2021-6-29

Strengths

1) the vanishing of the contribution of the inhomogeneous term to the ground-state energy is shown

2) new results for the surface energies are derived.

Weaknesses

1) the finite size correction term const.N^beta is not evaluated (analytically)

Report

The authors study the Zamolodchikov-Fateev model resp. the Takhtajan-Babujian model with spin-1 on a chain with open boundary conditions and boundary fields that break the conservation of magnetization.

Section 2 is devoted to the demonstration of integrability of the model by a compilation of results of the literature. This section ends with a presentation of the ODBA result for the eigenvalue of the transfer matrix with spin-1/2 representation in the auxiliary space and the fusion relation of this eigenvalue to the physically interesting one.

Section 3 is entitled "Finite size scaling behavior", but in fact it shows that a certain homogeneous approximation to the inhomogeneous T-Q relation yields the correct bulk O(N) and surface O(1) contributions to the energy. Technically, the reduction of the complete inhomogeneous T-Q relation to the homogeneous one (28) leads to homogeneous Bethe ansatz equations (29) which allow for an analytic solution in the thermodynamic limit which is presented in section 4. For the homogeneous approximation the formulas (32,35) are evaluated for system sizes up to N=200 and compared to the full results either by solving the inhomogeneous BAEs (24,25) directly or by density matrix renormalization group (DMRG) calculations. The difference is shown in figure 2 and is obviously an algebraically decaying function const.N^beta with negative values for beta in the neighbourhood of -1.

The authors should comment on the possibility of evaluating the finite size correction term const.N^beta by analytic means. For periodic boundary conditions the model of interest has been investigated long ago by non-linear integral equations

A. Kl"umper, M. T. Batchelor, P. A. Pearce: Central charges of the 6- and 19-vertex models with twisted boundary conditions J. Phys. A 24, 3111-3133 (1991)

J Suzuki: Spinons in magnetic chains of arbitrary spins at finite temperatures, J. Phys. A: Math. Gen. 32 2341 (1999)

Is a similar treatment conceivable for open boundary conditions?

In section 4 the formulas (32,35) are evaluated by integral equations for density functions and yield the (known) bulk energy and for the six identified regions explicit expressions for the surface energies are obtained.

In summary, the manuscript may qualify for publication in SciPost, because the vanishing of the contribution of the inhomogeneous term to the ground-state energy is shown and new results for the surface energies are derived. However the authors should respond satisfactorily to the above formulated question and the following questions/suggestions:

After (1) in line 28 the authors write "the SU(2) symmetry survives". I suggest to drop "survives" as model (1) is everywhere SU(2) invariant, just at the points J2/J1 = 1 and at J1=0 the model has higher SU(3) invariance.

In line 33 "J2 /J1 = 1/3, the Hamiltonian (1) degenerates into a projector operator" should be made more explicit: the "projector operator" is in fact the projection onto the sum of the spin-0 and spin-1 subspaces.

In line 82 the authors write "η is the crossing parameter", but here we are dealing with the isotropic model and there is no non-trivial crossing parameter. This parameter can be scaled out as is done in line 123. This should be indicated already in line 82.

In eq (6) and line 85 the operator P_12^0 is called antisymmetry but also "projector in the total spin-0 channel". What is really meant? The anti-symmetric states are the spin-1 states!

In line 107 "hierarchy fusion" should be called "fusion hierarchy".

On page 7 and later the terminology "boundary string" is somewhat confusing. It always refers to a single Bethe root.

The word "subfigure" should be replaced by "inset".

Line 213. "Meanwhile" should be replaced by a different word or simply by saying "Note that two holes λ^h1 and λ^h2 are introduced..."

spin long z-direction -> spin along the z-direction

Typo in caption to Fig.3: (68) pand the red points -> (68) and the red points

Requested changes

see above

  • validity: high
  • significance: good
  • originality: ok
  • clarity: good
  • formatting: good
  • grammar: reasonable

Author:  Xiaotian Xu  on 2021-08-12  [id 1660]

(in reply to Report 1 on 2021-06-29)

Thank you very much for your many helpful points raised about our paper and we really appreciate your support. We believe that these comments and suggestions have significantly improved our manuscript. To these comments we respond as follows. The page numbers and equation numbers refer to revised version, unless specify.

$~\textbf{1. Reviewer:}$ After (1) in line 28 the authors write "the $SU(2)$ symmetry survives". I suggest to drop "survives" as model (1) is everywhere $SU(2)$ invariant, just at the points $J_2/J_1=1$ and at $J_1=0$ the model has higher $SU(3)$ invariance.
$\quad~\textbf{Authors:}$ You are right. In the Hamiltonian (1), the spin-exchanging interaction in the bulk is $J_1\vec{S}_k\cdot \vec{S}_{k+1}+J_2(\vec{S}_k\cdot \vec{S}_{k+1})^2$, which has $SU(2)$ symmetries with arbitrary coupling constants. If $J_2/J_1=1$ or $J_1=0$, the model has the $SU(3)$ invariance. So we change the word "survives" by "exists".

$~\textbf{2. Reviewer:}$ In line 33 "$J_2/J_1=1/3$, the Hamiltonian (1) degenerates into a projector operator" should be made more explicit: the "projector operator" is in fact the projection onto the sum of the spin-0 and spin-1 subspaces.
$\quad~\textbf{Authors:}$ According to your suggestion, we adjust the explanation of projector operator. We modify the sentence in line 33 "the Hamiltonian (1) degenerates into a projector operator" into "the Hamiltonian (1) degenerates into a projector operator that is in fact the projection onto the sum of the spin-0 and spin-1 subspaces".

$~\textbf{3. Reviewer:}$ In line 82 the authors write "$\eta$ is the crossing parameter", but here we are dealing with the isotropic model and there is non-trivial crossing parameter. This parameter can be scaled out as is done in line 123. This should be indicated already in line 82.
$\quad~\textbf{Authors:}$ Thank you very much for your suggestion. In line 82, we have indicated that $\eta$ can be scaled out.

$~\textbf{4. Reviewer:}$ In eq(6) and line 85 the operator $\textbf{P}_{12}^{(0)}$ is called antisymmetry but also "projector in the total spin-0 channel". What is really meant? The anti-symmetric states are the spin-1 states!
$\quad~\textbf{Authors:}$ Thank you for your kind advice. At the point of $u=-\eta$, the $R$-matrix (2) is a singular matrix which only has one non-zero eigenvalue. Therefore, when the spectral parameter $u$ equals to $-\eta$, the $R$-matrix (2) degenerates into a one-dimensional projector operator $\textbf{P}_{12}^{(0)}=|\psi_0\rangle\langle\psi_0|$ and the corresponding basis vector is $|\psi_0\rangle=\frac{1}{\sqrt 3}(|1,-1\rangle-|0,0\rangle+|-1,1\rangle)$, where $|1\rangle$, $|0\rangle$ and $|-1\rangle$ are the eigenstates of spin-1 operator $S^z$ with the eigenvalues 1, 0 and -1, respectively. The operator $\textbf{P}_{12}^{(0)}$ projects the tensor space $\mathbb{V} \otimes \mathbb{V}$ into spin-0 subspace. We modified "Antisymmetry" to "Fusion condition" in Eq.(6).

$~\textbf{5. Reviewer:}$ In line 107 "hierarchy fusion" should be called "fusion hierarchy".
$\quad~\textbf{Authors:}$ Thank you for your careful reading. We have modified "hierarchy fusion" as "fusion hierarchy".

$~\textbf{6. Reviewer:}$ On page 7 and later the terminology "boundary string" is somewhat confusing. It always refers to a single Bethe root.
$\quad~\textbf{Authors:}$ The boundary strings mean that the values of Bethe roots are determined by the boundary parameters. They could be the single value or the bound pairs. For example, the boundary strings in regime II are {$pi, (p-1)i, -pi, -(p-1)i$}, which are clearly related with the boundary fields and distribute symmetrically around the zero point. Due to the symmetry, the boundary strings {$pi, (p-1)i$} and the {$-pi, -(p-1)i$} have the same contribution to the boundary energy.

$~\textbf{7. Reviewer:}$ The word "subfigure" should be replaced by "inset".
$\quad~\textbf{Authors:}$ We have replaced the word "subfigure" with "inset".

$~\textbf{8. Reviewer:}$ Line 213. "Meanwhile" should be replaced by a different word simply by saying "Note that two holes $\lambda_1^h$ and $\lambda_2^h$ are introduced $\cdots$"
$\quad~\textbf{Authors:}$ Thank you for your kind advice. We have replaced "Meanwhile, two holes $ \lambda_1^h $ and $ \lambda_2^h $ should be introduced" with "Note that two holes $\lambda_1^h$ and $\lambda_2^h$ are introduced".

$~\textbf{9. Reviewer:}$ spin long $z$-direction $\rightarrow$ spin along the $z$-direction.
$\quad~\textbf{Authors:}$ Thank you for your careful reading. We have replaced all the "spin long $z$-direction" with "spin along the $z$-direction".

$~\textbf{10. Reviewer:}$ Typo in caption to Fig.3: (68) pand the red points $\rightarrow$ (68) and the red points.
$\quad~\textbf{Authors:}$ Thank you for your careful reading. We have corrected the typo.

$~\textbf{11. Reviewer:}$ The authors should comment on the possibility of evaluating the finite size correction term const.$N^{\beta}$ by analytic means. For periodic boundary conditions the model of interest has been investigated long ago by non-linear integral equations.
A. Klumper, M. T. Batchelor, P. A. Pearce: Central charges of the 6- and 19-vertex models with twisted boundary conditions. J. Phys. A 24, 3111-3133 (1991).
J Suzuki: Spinons in magnetic chains of arbitrary spins at finite temperatures, J. Phys. A: Math. Gen. 32 2341 (1999).
Is a similar treatment conceivable for open boundary conditions?
$\quad~\textbf{Authors:}$ Very good suggestion! The finite size correction of quantum integrable system is an interesting issue. Based on the logarithmic and integral forms of Bethe ansatz equations, some physical quantities such as the long-range correlation function, elementary excitation, dressed energy, ground state energy correction, susceptibility and Drude weight can be calculated. In the references mentioned by the referee, the results of some models with periodic boundary condition have been obtained. We think that this method is also valid for the open boundary conditions, if the Bethe ansatz equations are homogeneous. Because the starting points of this method are the energy spectrum and Bethe ansatz equations, and the processes of treating are the same. Obviously, the finial physical properties could be different.
$\quad$Focus on the present model (15). Starting from the reduced Bethe ansatz equation (29), we can analytically compute the finite size corrections and obtain the approximate solutions. The more accurate solutions should be obtained from the Bethe ansatz equations (24). However, due to the existence of inhomogeneous terms in Eq.(24), it is hard to analytically calculate the finite size correction. We think that this difficulty could be overcome by using the recently proposed $t-W$ relation [Phys. Rev. B 102, 085115 (2020)] or the fusion hierarchy at the special points of $u=\{\theta_k,k=1,\cdots,N\}$ [Phys. Rev. B 103, L220401 (2021)]. The main idea is that the eigenvalue of transfer matrix $t(u)$ can also be characterized by its zero points, instead of the Bethe roots. The zero points satisfy the homogeneous Bethe ansatz equations. Based on them, we can analytically calculate the finite size correction.

---

## Round 1 · Referee Report · Anonymous (Referee 2) · 2021-7-7

Strengths

1- New results on the dependence of the surface energy of the integrable spin-1 Heisenberg model on (diagonal!) boundary fields

Weaknesses

1- The effect of non-diagonal boundary fields to the ground state energy is of order $O(L^{-1})$ and therefore not captured in the "reduced" TQ- or Bethe equations.

Report

The authors study the bulk and surface contributions to the ground state energy of the integrable spin-1 Heisenberg chain with open boundary conditions. Non-parallel boundary fields break the $U(1)$ symmetry of the system and a complete solution requires the use of the so-called off-diagonal Bethe ansatz (ODBA) based on an inhomogeneous extension of the TQ-equation.

The ODBA solution from Ref. [40] leading to the inhomogeneous TQ-equation (17,18) for the transfer matrix eigenvalues is presented in Section 2. (NB: the general off-diagonal boundary matrices should be contain 3 parameters for strenght and direction).

In Section 3 this equation is approximated to give a "reduced TQ-relation" which, in fact, are nothing but the (homogeneous) eigenvalue equation obtained by means of the algebraic Bethe ansatz for parallel boundary fields (which can be seen by comparing (27)-(29) with, e.g., the rational limit of the corresponding equations in Ref. [26]). The boundary parameters $p$ and $q$ appearing in the reduced equations are the combinations of the original non-diagonal parameters obtained by rotation of the boundary matrices (see Ref. [29]).

From their analysis of the "reduced" Bethe equations (29) the authors therefore obtain the bulk $\sim O(L^1)$ and surface energy $\sim O(L^0)$ of the spin-1 model as function of (diagonal) boundary fields parameterized by $p$ and $q$. In the thermodynamic limit the boundary contributions $\sim O(L^0)$ from the two ends of the chain decouple, therefore it is not surprising that these quantities coincide with the numerical results obtained from the full ODBA solution or DRMG calculation for non-parallel fields. The effect of the latter will only show up in a true finite size scaling analysis, i.e. considering
$O(L^{-1})$ corrections.

The identification of root configurations to the reduced Bethe equations and the calculation of the boundary contribution (68) to the ground state energy as function of the diagonal boundary fields $p$ and $q$, however, is new (to my knowledge). Putting this into the context of the more general model and its solution by means of the ODBA, however, is misleading.

A revision focussed on the boundary energy of the model with parallel fields is worth to be published, although it might be more suitable for SciPost Physics Core.
  • validity: good
  • significance: good
  • originality: ok
  • clarity: good
  • formatting: good
  • grammar: good

Author:  Xiaotian Xu  on 2021-08-12  [id 1661]

(in reply to Report 2 on 2021-07-07)

Thank you very much for your many helpful points raised about our paper and we really appreciate your support. We believe that these comments and suggestions have significantly improved our manuscript. To these comments we respond as follows. The page numbers and equation numbers refer to revised version, unless specify.

  1. Reviewer: In Section 3 this equation is approximated to give a "reduced TQ-relation" which, in fact, are nothing but the (homogeneous) eigenvalue equation obtained by means of the algebraic Bethe ansatz for parallel boundary fields (which can be seen by comparing (27)-(29) with, e.g., the rational limit of the corresponding equations in Ref. [26]). The boundary parameters p and q appearing in the reduced equations are the combinations of the original non-diagonal parameters obtained by rotation of the boundary matrices (see Ref. [29]). The identification of root configurations to the reduced Bethe equations and the calculations of the boundary contribution (68) to the ground state energy as function of the diagonal boundary fields $p$ and $q$, however, is new (to my knowledge). Putting this into the context of the more general model and its solution by means of the ODBA, however, is misleading. Authors: Thank you very much! The motivation of this paper is that we want to calculate the physical quantities such as the boundary energy of the spin-1 Heisenberg chain induced by the unparallel boundary magnetic fields. The exact energy spectrum of the system is characterized by the inhomogeneous $T-Q$ relations (17)-(23) and associated Bethe ansatz equations (24) for any finite $N$. Due to the existence of inhomogeneous terms in Eq.(24) induced by the unparallel boundary fields, the traditional thermodynamic Bethe asnatz method does not work. By using the density matrix renormalization group simulation, we find that the contribution of inhomogeneous terms in the $T-Q$ relations can be neglected at the ground state only in the thermodynamic limit. Namely, one of most important results of this paper is that we show here the inhomogeneous $T-Q$ relation (17)-(23) and Bethe ansatz equations (24) for the spin-1 Heisenberg chain with the unparallel boundary fields, only in the thermodynamic limit, can be simulated by the homogeneous ones (27)-(29) (which are equivalent to those with diagonal boundary fields). The physical picture is that the two boundary fields can not "see" each other only in the thermodynamic limit (i.e., the contributions to the boundary energy of the two boundary fields independently). [We shall note that the eigenstates with diagonal and those with non-diagonal boundary fields are quite different.] From this equivalent diagonal boundary fields (only in thermodynamic limit), we then obtain the analytical expression for the boundary energy. We note that the boundary energy (68) includes the contribution of off-diagonal boundary parameters $\alpha_{\pm}$. All the results are reasonable and consistent with each other.

  2. Reviewer: The ODBA solution from Ref.[40] leading to the inhomogeneous TQ-equation (17,18) for the transfer matrix eigenvalues is presented in Section 2. (NB: the general off-diagonal boundary matrices should be contain 3 parameters for strength and direction). Authors: Thank you very much for pointing the fact that the general off-diagonal boundary matrices should be contain 3 parameters for strength and direction. We use the general off-diagonal boundary matrix $K^-(u)$ in (7)-(8) with the boundary parameters ${p_-,\,\alpha_-,\phi_-}$ characterizing the strength and direction of the boundary field (resp. $K^+(u)$ in (10) with the boundary parameters ${p_+,\,\alpha_+,\phi_+}$ ). The eigenvalues $\Lambda(u)$ of the transfer matrix is given by the inhomogeneous $T-Q$ relations (17)-(23) for any finite $N$, which implies that the energy of the ground state may depend on the boundary parameters through the values: $p_{\pm},\,\alpha_{\pm}$ and $\phi_--\phi_+$. However, in the thermodynamic limit, the boundary energy only depends on the boundary parameters through the values: $p=\frac{p_{-}}{\sqrt{1+\alpha^2_{-}}}-\frac{1}{2}$ and $q=-\frac{p_{+}}{\sqrt{1+\alpha^2_{+}}}-\frac{1}{2}$ (see (63), (67) and (68)).

---

## Round 1 · Referee Report · Anonymous (Referee 3) · 2021-7-22

Strengths

Some new interesting results.

Weaknesses

The main results derived are for boundary conditions different from those declared from the author, the numerical analysis developed is mainly stated without presenting in the manuscript enough data

Report

The manuscript presents some interesting and apparently new results on the surface energy of the open integrable spin-1 Heisenberg chain while the authors also claim to have derived some interesting numerical results for large chains. The presentation of the material is however strongly misleading if not even incorrect, leading to state analytic results not for the proper boundary conditions. I found moreover, that the manuscript is rather scarce in the citation of the relevant literature and too much centered on the previous production of the authors. It is my opinion that the manuscript needs a detailed rewriting to take into account the critics of the previous referees as well as those that I will detail in the following. After that, one can evaluate if the manuscript fulfills the minimal requirement for the publication on SciPost or, as suggested by the second referee, a sub-thematic one.

I will focus my comments on the presentation of the material and the results in the manuscript. The authors dedicate the section 2 to introduce the open spin-1 XXX chain with some general boundary conditions and then they recall the TQ-equations for the transfer matrix eigenvalues according to their ODBA Ansatz method. These TQ-equations contains an “inhomogeneous term” the thermodynamic analysis of the associated Bethe equations is not known currently and so the authors implement this analysis for the Bethe equations associated to the homogeneous TQ-equation. They call this the “reduced Bethe equation”, but in fact they are just the Bethe equations of the same open spin-1 XXX chain for different boundary conditions. This changes the boundary conditions in “diagonal ones”. This changes the u (spectral parameter) asymptotic behavior of the transfer matrix allowing to the eigenvalues of the (1/2,1) transfer matrix to satisfy the homogeneous TQ-equation. Then, the analysis developed in section 3 and 4 on the Bethe roots and the boundary energy in the thermodynamic limit, in fact, are for these “diagonal boundary conditions” and not for the un non-diagonal boundary magnetic fields case. Indeed, in section 4, for all the considered regimes I-IV, they always refer to the Bethe equations associated to the homogeneous TQ-equations. This is very different from what the authors write already in the title and in their abstract, see also for example the statements at the beginning of section 4:
“In this section, we study the physical effects induced by the unparallel boundary magnetic fields and compute the boundary energy”.
For the non-diagonal case, the authors have just the numerical analysis by DRMG that they claim to have developed for long chains up to 200 sites. In fig. 2, the result of this numerical analysis is the convergence like N^{\beta}, with \beta around -1, of the ground state energy density for the non-diagonal and diagonal one. In fig.3, they claim to have extrapolated the thermodynamic limit of their DMRG results of the ground state boundary energy for the non-diagonal case and shown the coincidence with the explicit functions (68) found for the diagonal case. It would be nice and useful to have more detailed data of their DMRG analysis and a figure, for each chosen values of p and q for which they have done their computations. In particular, they should plot the DMRG results for the chain of N=4, 14, …,194 and from which the asymptotic values are extrapolated. Let me say, that while I am not a specialist of numerical analysis, like DMRG, and so I am unable to verify even for shorter chains the numerical claims of the authors what they present looks reasonable from a physical point of view. However, one should stress that the thermodynamic convergence of the ground state boundary energy density of the non-diagonal and diagonal cases does not imply that in general one can just consider the “reduced” Bethe equations to develop the thermodynamic analysis of the relevant physical quantities in the non-diagonal case. This has been stressed already by the second referee and I agree with the first one on the relevance of having an analytic derivation of the \beta as functions of the boundary parameters. For example, this type of finite size analysis can be important for relevant physical quantities such as the correlation functions where a modification of order 1/N in the ground state Bethe root density can produce a modification of the correlation functions; e.g. this is clearly the case if one thinks to the exact results for the XXZ spin 1/2 chain.
The authors should also look for misprints, e.g. in “The model Hamiltonian is generalized from the transfer matrix t(u) as” it is “generated” and not “generalized”. Then about the Hamiltonian, there are more strange things. The definition of H in terms of the transfer matrix in the first line of (15) is apparently singular in u=0. Moreover, it produces a term which is the logarithmic derivative of the transfer matrix which is not the standard definition for the open chain. There is some strange asymmetry between the expressions of the boundary magnetic fields in the site 1 and site N with respect to the boundary parameters minus and plus. A part an overall 1/\eta in the site N there is also a different behavior with respect to the boundary parameters \alpha_{-} and \alpha_{+}. One expects that when these two values are zero one gets back the diagonal case, i.e., the two boundary fields parallel and oriented along the z-direction. However, for \alpha_{-}=0 the boundary magnetic field in the site 1 become oriented along the z-direction while this is not the case for \alpha_{+}=0 for the boundary magnetic field in the site N. Even more strange, with the current parametrization, the boundary magnetic field in the site N seems to be orientable in the z-direction only for specific values of both the boundary parameters, e.g. (\alpha{+}=0,p_{+}^2=3\eta^2/4). If there are no mistakes the authors should comment why from a symmetric writing of the transfer matrix with respect to the boundary matrices (and so the boundary parameters) one gets such asymmetric Hamiltonian.

Requested changes

See above

  • validity: ok
  • significance: ok
  • originality: ok
  • clarity: poor
  • formatting: reasonable
  • grammar: reasonable

Author:  Xiaotian Xu  on 2021-08-12  [id 1662]

(in reply to Report 3 on 2021-07-22)

Thank you very much for your many helpful points raised about our paper and we really appreciate your support. We believe that these comments and suggestions have significantly improved our manuscript. To these comments we respond as follows. The page numbers and equation numbers refer to revised version, unless specify.

  1. Reviewer: The main results derived are for boundary conditions different from those declared from the author. $\quad$The manuscript presents some interesting and apparently new results on the surface energy of the open integrable spin-1 Heisenberg chain while the authors also claim to have derived some interesting numerical results for large chains. The presentation of the material is however strongly misleading if not even incorrect, leading to state analytic results not for the proper boundary conditions. $\quad$They call this the "reduced Bethe equation", but in fact they are just the Bethe equations of the same open spin-1 XXX chain for different boundary conditions. This changes the boundary conditions in "diagonal ones". This changes the u (spectral parameter) asymptotic behavior of the transfer matrix allowing to the eigenvalues of the (1/2,1) transfer matrix to satisfy the homogeneous TQ-equation. Then, the analysis developed in section 3 and 4 on the Bethe roots and the boundary energy in the thermodynamic limit, in fact, are for these "diagonal boundary conditions" and not for the un non-diagonal boundary magnetic fields case. Indeed, in section 4, for all the considered regimes I-IV, they always refer to the Bethe equations associated to the homogeneous TQ-equations. This is very different from what the authors write already in the title and in their abstract, see also for example the statements at the beginning of section 4: “In this section, we study the physical effects induced by the unparallel boundary magnetic fields and compute the boundary energy”. Authors: The motivation of this paper is that we want to calculate the physical quantities such as the boundary energy of the spin-1 Heisenberg chain induced by the unparallel boundary magnetic fields. The exact energy spectrum of the system is characterized by the inhomogeneous $T-Q$ relations (17)-(23) and associated Bethe ansatz equations (24) for any finite $N$. Due to the existence of inhomogeneous terms in Eq.(24) induced by the unparallel boundary fields, the traditional thermodynamic Bethe asnatz method does not work. By using the density matrix renormalization group simulation, we find that the contribution of inhomogeneous terms in the $T-Q$ relations can be neglected at the ground state only in the thermodynamic limit. Namely, one of most important results of this paper is that we show here the inhomogeneous $T-Q$ relation (17)-(23) and Bethe ansatz equations (24) for the spin-1 Heisenberg chain with the unparallel boundary fields, only in the thermodynamic limit, can be simulated by the homogeneous ones (27)-(29) (which are equivalent to those with diagonal boundary fields). The physical picture is that the two boundary fields can not "see" each other only in the thermodynamic limit (i.e., the contributions to the boundary energy of the two boundary fields independently). [We shall note that the eigenstates with diagonal and those with non-diagonal boundary fields are quite different.] From this equivalent diagonal boundary fields (only in thermodynamic limit), we then obtain the analytical expression for the boundary energy. We note that the boundary energy (68) includes the contribution of off-diagonal boundary parameters $\alpha_{\pm}$. All the results are reasonable and consistent with each other.

  2. Reviewer: I found moreover, that the manuscript is rather scarce in the citation of the relevant literature and too much centered on the previous production of the authors. Authors: Thank you for your kind advice. We deleted some self-citations and added three new references: [33] J. Cao, W.-L. Yang, K. Shi and Y. Wang, Off-diagonal Bethe ansatz solution of the XXX spin-chain with arbitrary boundary conditions, Nucl. Phys. B 875, 152 (2013). [34] R.I. Nepomechie, An inhomogeneous $T-Q$ equation for the open XXX chain with general boundary terms: completeness and arbitrary spin, J. Phys. A: Math. Theor. 46, 442002 (2013). [35] R.I. Nepomechie and C. Wang, Boundary energy of the open XXX chain with a non-diagonal boundary term, J. Phys. A: Math. Theor. 47, 032001 (2014).

  3. Reviewer: However, one should stress that the thermodynamic convergence of the ground state boundary energy density of the non-diagonal and diagonal cases does not imply that in general one can just consider the "reduced" Bethe equations to develop the thermodynamic analysis of the relevant physical quantities in the non-diagonal case. This has been stressed already by the second referee and I agree with the first one on the relevance of having an analytic derivation of the $\beta$ as functions of the boundary parameters. For example, this type of finite size analysis can be important for relevant physical quantities such as the correlation functions where a modification of order 1/N in the ground state Bethe root density can produce a modification of the correlation functions; e.g. this is clearly the case if one thinks to the exact results for the XXZ spin 1/2 chain. Authors: Very good suggestion! The finite size correction of quantum integrable system is an interesting issue. Based on the logarithmic and integral forms of Bethe ansatz equations, some physical quantities such as the long-range correlation function, elementary excitation, dressed energy, ground state energy correction, susceptibility and Drude weight can be calculated. In the references [A. Klumper, M. T. Batchelor and P. A. Pearce, Central charges of the 6- and 19-vertex models with twisted boundary conditions, J. Phys. A 24, 3111 (1991); J Suzuki, Spinons in magnetic chains of arbitrary spins at finite temperatures, J. Phys. A: Math. Gen. 32, 2341 (1999)], the results of some models with periodic boundary condition have been obtained. We think that this method is also valid for the open boundary conditions, if the Bethe ansatz equations are homogeneous. Because the starting points of this method are the energy spectrum and Bethe ansatz equations, and the processes of treating are the same. Obviously, the finial physical properties could be different. $\quad$Focus on the present model (15). Starting from the reduced Bethe ansatz equation (29), we can analytically compute the finite size corrections and obtain the approximate solutions. The more accurate solutions should be obtained from the Bethe ansatz equations (24). However, due to the existence of inhomogeneous terms in Eq.(24), it is hard to analytically calculate the finite size correction. We think that this difficulty could be overcome by using the recently proposed $t-W$ relation [Phys. Rev. B 102, 085115 (2020)] or the fusion hierarchy at the special points of $u={\theta_k,k=1,\cdots,N}$ [Phys. Rev. B 103, L220401 (2021)]. The main idea is that the eigenvalue of transfer matrix $t(u)$ can also be characterized by its zero points, instead of the Bethe roots. The zero points satisfy the homogeneous Bethe ansatz equations. Based on them, we can analytically calculate the finite size correction.

  4. Reviewer: The numerical analysis developed is mainly stated without presenting in the manuscript enough data. $\quad$It would be nice and useful to have more detailed data of their DMRG analysis and a figure, for each chosen values of $p$ and $q$ for which they have done their computations. In particular, they should plot the DMRG results for the chain of N=4, 14, …,194 and from which the asymptotic values are extrapolated. Authors: Very good suggestion! According to your suggestion, we have added the Figure 4 to show the DMRG analysis. The corresponding description in the manuscript is also revised. $\quad$In order to check the correction of analytical expression of boundary energy (68), we perform the numerical simulation with DMRG algorithm. The ground state energy $E_g(N)$ of the model (15) with finite system size $N$ can be calculated by using the DMRG method. Then we consider the physical quantity

    $$ e_b(N) =E_g(N)-Ne_g,\tag{1} $$
    where $e_g=-1$ is the ground state energy density of the system with periodic boundary condition [Please see Eq.(55) in the manuscript]. Obviously, in the thermodynamic limit, the value of $e_b(N\rightarrow\infty)$ gives the boundary energy. $\quad$In the following Figs.1-4 (please see attachment), we show the numerical results. The quantity $e_b(N)$ is calculated with the system size $N = 10(n-1)+4$ and $n=1,2,\cdots, 20$. The boundary parameters $p$ and $q$ take the values as those given by Fig.3 in the manuscript. The red points are the numerical values of $e_b(N)$, the blue solid lines are the fitting curves, and the red solid lines are the extrapolated boundary energies. From the data, we find that the $e_b(N)$ and $N$ satisfy the power law relation, i.e., $e_b(N)=aN^b+c$. Due to the fact that $b<0$, in the thermodynamic limit, the asymptotic value $c$ determines the boundary energy. Comparing $c$ with the analytical result (68), we find that all the results agree with each other very well. $~$

  5. Reviewer: In "The model Hamiltonian is generalized from the transfer matrix $t(u)$ as" it is "generated" and not "generalized". Authors: Thank you for your careful reading. We have replaced the word "generalized" with "generated". $~$
  6. Reviewer: The definition of $H$ in terms of the transfer matrix in the first line of (15) is apparently singular in $u=0$. Moreover, it produces a term which is the logarithmic derivative of the transfer matrix which is not the standard definition for the open chain. Authors: Thank you very much for pointing out our misprints. We have corrected the misprint of the definition of the Hamiltonian (15). $~$
  7. Reviewer: There is some strange asymmetry between the expressions of the boundary magnetic fields in the site 1 and site $N$ with respect to the boundary parameters minus and plus. A part an overall $1/\eta$ in the site $N$ there is also a different behavior with respect to the boundary parameters $\alpha_{-}$ and $\alpha_{+}$. One expects that when these two values are zero one gets back the diagonal case, i.e., the two boundary fields parallel and oriented along the $z$-direction. However, for $\alpha_{-}=0$ the boundary magnetic field in the site 1 become oriented along the $z$-direction while this is not the case for $\alpha_{+}=0$ for the boundary magnetic field in the site $N$. Even more strange, with the current parametrization, the boundary magnetic field in the site $N$ seems to be orientable in the $z$-direction only for specific values of both the boundary parameters, e.g. ($\alpha{+}=0,p_{+}^2=3\eta^2/4$). If there are no mistakes the authors should comment why from a symmetric writing of the transfer matrix with respect to the boundary matrices (and so the boundary parameters) one gets such asymmetric Hamiltonian. Authors: When the non-diagonal boundary parameters $\alpha_\pm=0$, the Hamiltonian reads
    $$ \begin{eqnarray} H&=&\frac{1}{\eta}\sum_{k=1}^{N-1} \left[\vec S_{k} \cdot \vec S_{k+1} -(\vec S_{k} \cdot \vec S_{k+1})^2 \right] + \frac{1}{p_-^2-\frac{1}{4}\eta^2}\left[ 2p_-S_{1}^z -\eta (S_{1}^z )^2 \right] \ && +\frac{1}{(p_+^2-\frac{1}{4}\eta^2)\eta} \left[ -2p_+ \eta S_{N}^z -\frac{2}{3}p_+^2 [(S_{N}^x )^2+(S_{N}^y )^2+(S_{N}^z )^2] +\frac{\eta^2}{2}[ (S_{N}^x )^2+(S_{N}^y )^2-(S_{N}^z )^2 ]\right] \ && +\frac{\eta}{3p_+^2-\frac{3}{4}\eta^2} +\frac{\eta}{p_-^2-\frac{1}{4}\eta^2}+\frac1{\eta}3N+\frac{4}{\eta} \tag{2} \ &=&\frac{1}{\eta}\sum_{k=1}^{N-1} \left[\vec S_{k} \cdot \vec S_{k+1} -(\vec S_{k} \cdot \vec S_{k+1})^2 \right] + \frac{1}{p_-^2-\frac{1}{4}\eta^2}\left[ 2p_-S_{1}^z -\eta (S_{1}^z )^2 \right] \ && +\frac{1}{p_+^2-\frac{1}{4}\eta^2} \left[ -2p_+ S_{N}^z -\eta (S_{N}^z )^2\right] \ && +\frac{\eta}{p_-^2-\frac{1}{4}\eta^2} + \frac{4\eta^2-4p_+^2}{(3p_+^2-\frac{3}{4}\eta^2)\eta}+\frac1{\eta}3N+\frac{4}{\eta},\tag{3} \end{eqnarray} $$
    where we have used the identity $(S^x)^2+(S^y)^2+(S^z)^2=2$. We see that in the fused Hamiltonian $(3)$, the boundary fields are indeed parallel and both of them are along the $z$-direction. Putting $\eta=1$ and changing the signs of boundary parameters at one side, the Hamiltonian $(3)$ can also be written into the symmetric form. The asymmetry of the Hamiltonian $(2)$ is apparent.

Attachment:

Figs.1-4.pdf

---

## Round 2 · Referee Report · Anonymous (Referee 1) · 2021-8-31

Strengths

1) the vanishing of the contribution of the inhomogeneous term to the ground-state energy is shown

2) new results for the surface energies are derived.

Weaknesses

1) the finite size correction term const.N^beta is not evaluated (analytically)

Report

The authors have improved their manuscript. I therefore recommend publication
of the manuscript.

Please write (15) in a more symmetric form where the left and the right
boundary terms appear on equal footing.

Figure 4: please change the exponent b to \beta as in Figure 2.

The discussion should contain information on the status of the finite-size
analysis:

-- the O(N^1) bulk term and the O(N^0) boundary terms to the ground state
energy do not depend on the orientation of the boundary fields,

-- the true finite size terms are probably of order O(N^-1) and are out of
reach for the inhomogeneous/off-diagonal case,

-- due to higher order terms the effective exponents determined in the
manuscript differ somewhat from -1

and maybe

-- the diagonal case is possibly tractable along the lines of A. Klumper et
al. and J Suzuki

Please remove as many language problems as possible. There are a couple of
them.
  • validity: high
  • significance: good
  • originality: ok
  • clarity: good
  • formatting: good
  • grammar: reasonable

Author:  Xiaotian Xu  on 2021-10-14  [id 1853]

(in reply to Report 1 on 2021-08-31)

Thank you very much for your many helpful points raised about our paper and we really appreciate your support. We believe that these comments and suggestions have significantly improved our manuscript. To these comments we respond as follows. The page numbers and equation numbers refer to revised version, unless specify.

1) Reviewer: Please write (15) in a more symmetric form where the left and the right boundary terms appear on equal footing.

Authors: Thank you for your suggestion. We rewrote the Hamiltonian (15) in a more symmetric form.

2) Reviewer: Figure 4: please change the exponent b to $\beta$ as in Figure 2.

Authors: Thank you for your careful reading. We changed the exponent "b" to "$\beta$" in caption of Figure 4 and lines 297, 298.

3) Reviewer: The discussion should contain information on the status of the finite-size analysis:

-- the $O(N^1)$ bulk term and the $O(N^0)$ boundary terms to the ground state energy do not depend on the orientation of the boundary fields,

-- the true finite size terms are probably of order $O(N^{-1})$ and are out of reach for the inhomogeneous/off-diagonal case,

-- due to higher order terms the effective exponents determined in the manuscript differ somewhat from -1.

and maybe

-- the diagonal case is possibly tractable along the lines of A. Klumper et al. and J Suzuki.

Authors: We added the discussion at the end of section 3 in P.10, and added three references [45], [46] and [47] mentioned in the report.

Due to the existence of inhomogeneous term in BAEs.(24), it is hard to analytically calculate the finite size correction for the present off-diagonal boundary reflections along the lines given in references [45-47]. We shall note that the diagonal case is tractable along the lines of A. Klumper et al. [46] and J. Suzuki [47]. The finite size correction $\mathcal{O}(N^1)$ for the bulk and $\mathcal{O}(N^0)$ term for the boundaries to the ground state energy do not depend on the orientations of the boundary fields. The true finite size correction terms are probably of order $\mathcal{O}(N^{-1})$ and are out of reach for the inhomogeneous/off-diagonal case. Due to higher order correction terms, the effective exponents $\beta$ determined in the paper differ from $-1$.

4) Reviewer: Please remove as many language problems as possible. There are a couple of them.

Authors: Thank you very much for your careful reading. We have corrected the language problems and polished the English.

Author:  Xiaotian Xu  on 2021-10-14  [id 1849]

(in reply to Report 1 on 2021-08-31)

Thank you very much for your many helpful points raised about our paper and we really appreciate your support. We believe that these comments and suggestions have significantly improved our manuscript. To these comments we respond as follows. The page numbers and equation numbers refer to revised version, unless specify. 1. Reviewer: Please write (15) in a more symmetric form where the left and the right boundary terms appear on equal footing. Authors: Thank you for your suggestion. We rewrote the Hamiltonian (15) in a more symmetric form. 2. Reviewer: Figure 4: please change the exponent b to $\beta$ as in Figure 2. Authors: Thank you for your careful reading. We changed the exponent "b" to "$\beta$" in caption of Figure 4 and lines 297, 298. 3. Reviewer: The discussion should contain information on the status of the finite-size analysis: -- the $O(N^1)$ bulk term and the $O(N^0)$ boundary terms to the ground state energy do not depend on the orientation of the boundary fields, -- the true finite size terms are probably of order $O(N^{-1})$ and are out of reach for the inhomogeneous/off-diagonal case, -- due to higher order terms the effective exponents determined in the manuscript differ somewhat from -1. and maybe -- the diagonal case is possibly tractable along the lines of A. Klumper et al. and J Suzuki. Authors: We added the discussion at the end of section 3 in P.10, and added three references [45], [46] and [47] mentioned in the report. Due to the existence of inhomogeneous term in BAEs.(24), it is hard to analytically calculate the finite size correction for the present off-diagonal boundary reflections along the lines given in references [45-47]. We shall note that the diagonal case is tractable along the lines of A. Klumper et al. [46] and J. Suzuki [47]. The finite size correction $\mathcal{O}(N^1)$ for the bulk and $\mathcal{O}(N^0)$ term for the boundaries to the ground state energy do not depend on the orientations of the boundary fields. The true finite size correction terms are probably of order $\mathcal{O}(N^{-1})$ and are out of reach for the inhomogeneous/off-diagonal case. Due to higher order correction terms, the effective exponents $\beta$ determined in the paper differ from $-1$. 4. Reviewer: Please remove as many language problems as possible. There are a couple of them. Authors: Thank you very much for your careful reading. We have corrected the language problems and polished the English.

Author:  Xiaotian Xu  on 2021-10-14  [id 1847]

(in reply to Report 1 on 2021-08-31)

Thank you very much for your many helpful points raised about our paper and we really appreciate your support. We believe that these comments and suggestions have significantly improved our manuscript. To these comments we respond as follows. The page numbers and equation numbers refer to revised version, unless specify.

  1. Reviewer: Please write (15) in a more symmetric form where the left and the right boundary terms appear on equal footing.

Authors: Thank you for your suggestion. We rewrote the Hamiltonian (15) in a more symmetric form.

  1. Reviewer: Figure 4: please change the exponent b to $\beta$ as in Figure 2.

Authors: Thank you for your careful reading. We changed the exponent "b" to "$\beta$" in caption of Figure 4 and lines 297, 298.

  1. Reviewer: The discussion should contain information on the status of the finite-size analysis:

-- the $O(N^1)$ bulk term and the $O(N^0)$ boundary terms to the ground state energy do not depend on the orientation of the boundary fields,

-- the true finite size terms are probably of order $O(N^{-1})$ and are out of reach for the inhomogeneous/off-diagonal case,

-- due to higher order terms the effective exponents determined in the manuscript differ somewhat from -1.

and maybe

-- the diagonal case is possibly tractable along the lines of A. Klumper et al. and J Suzuki.

Authors: We added the discussion at the end of section 3 in P.10, and added three references [45], [46] and [47] mentioned in the report.

Due to the existence of inhomogeneous term in BAEs.(24), it is hard to analytically calculate the finite size correction for the present off-diagonal boundary reflections along the lines given in references [45-47]. We shall note that the diagonal case is tractable along the lines of A. Klumper et al. [46] and J. Suzuki [47]. The finite size correction $\mathcal{O}(N^1)$ for the bulk and $\mathcal{O}(N^0)$ term for the boundaries to the ground state energy do not depend on the orientations of the boundary fields. The true finite size correction terms are probably of order $\mathcal{O}(N^{-1})$ and are out of reach for the inhomogeneous/off-diagonal case. Due to higher order correction terms, the effective exponents $\beta$ determined in the paper differ from $-1$.

  1. Reviewer: Please remove as many language problems as possible. There are a couple of them.

Authors: Thank you very much for your careful reading. We have corrected the language problems and polished the English.

---

## Round 2 · Referee Report · Anonymous (Referee 2) · 2021-9-3

Report

In their revision the authors have taken into account the technical points raised in the referees' reports. With respect to their numerical work they have added Fig.~4 showing that the $O(N^0)$ contribution to the ground state energy of the system converges to that computed from the Bethe ansatz equations (31), (32). Note, however, that the latter are based on the string hypothesis (i.e. strings with exponential accuracy of their components) which is known not to capture the $O(1/N)$ contributions to the energy correctly.
Still, this gives further support to their finding that bulk and boundary energy are correctly obtained using established Bethe ansatz techniques - even for non-parallel boundary fields.Together with the explicit calculation of the boundary energy of the integrable spin-1 chain this is a result which is worth to be published in SciPost Physics.

The referees' objections concerning the partly ambiguous discussion of the nature of boundary conditions considered in the paper, however, have not been addressed satisfactorily.
Specifically, I would suggest changes along the following lines

Abstract, lines 5,6 (and similar in the Conclusion, lines 293-296):
... are analyzed. Based on our findings the boundary energy of the system in the thermodynamic limit can be obtained from Bethe ansatz equations (BAEs) of a related model with parallel boundary fields. These results ...

Introduction, line 67: delete "unparallel"

Section 3, lines 134-137: change to
... is not the eigenvalue $\Lambda(u)$ for any finite $N$ but rather that of the transfer matrix with parallel boundary fields of the same strength. In the limit $N\to\infty$ it will give, however, the correct boundary energy (...

Section 4, line 216/7:
delete "unparallel" and add "in the thermodynamic limit" at the end of the first sentence.

With such changes implemented the paper will meet the general acceptance criteria of SciPost Physics.
  • validity: -
  • significance: -
  • originality: -
  • clarity: -
  • formatting: -
  • grammar: -

Author:  Xiaotian Xu  on 2021-10-14  [id 1852]

(in reply to Report 2 on 2021-09-03)

Thank you very much for your many helpful points raised about our paper and we really appreciate your support. We believe that these comments and suggestions have significantly improved our manuscript. To these comments we respond as follows. The page numbers and equation numbers refer to revised version, unless specify.

1) Reviewer: Abstract, lines 5,6 (and similar in the Conclusion, lines 293-296): ... are analyzed. Based on our findings the boundary energy of the system in the thermodynamic limit can be obtained from Bethe ansatz equations (BAEs) of a related model with parallel boundary fields. These results ...

Authors: According to your suggestion, we modified the sentence lines 5-8 in Abstract "Based on the reduced Bethe ansatz equations (BAEs), we obtain the boundary energy of the system. These results ..." into "Based on our findings, the boundary energy of the system in the thermodynamic limit can be obtained from Bethe ansatz equations of a related model with parallel boundary fields. These results ...".

2) Reviewer: Introduction, line 67: delete "unparallel"

Authors: Thank you for your kind advice. We deleted the word "unparallel".

3) Reviewer: Section 3, lines 134-137: change to ... is not the eigenvalue $\Lambda(u)$ for any finite $N$ but rather that of the transfer matrix with parallel boundary fields of the same strength. In the limit $N\rightarrow\infty$ it will give, however, the correct boundary energy ...

Authors: According to your advice, we changed the sentence "the $\Lambda_{hom}(u)$ is not the eigenvalue $\Lambda(u)$ for any finite... of the paper)." in lines 138-141 of P. 6 to "the $ \Lambda_{hom}(u) $ is not the eigenvalue $\Lambda(u)$ for any finite $N$ but rather that of the transfer matrix with parallel boundary fields of the same strength. In the limit $N\rightarrow\infty$ it will give, however, the correct boundary energy (see the following parts of the paper)."

4) Reviewer: Section 4, line 216/7: delete "unparallel" and add "in the thermodynamic limit" at the end of the first sentence.

Authors: Thank you for your kind advice. We deleted the word "unparallel" in line 228 and added "in the thermodynamic limit" at the end of the first sentence in lines 228-229 of P.10.

Author:  Xiaotian Xu  on 2021-10-14  [id 1848]

(in reply to Report 2 on 2021-09-03)

Thank you very much for your many helpful points raised about our paper and we really appreciate your support. We believe that these comments and suggestions have significantly improved our manuscript. To these comments we respond as follows. The page numbers and equation numbers refer to revised version, unless specify. 1. Reviewer: Abstract, lines 5,6 (and similar in the Conclusion, lines 293-296): ... are analyzed. Based on our findings the boundary energy of the system in the thermodynamic limit can be obtained from Bethe ansatz equations (BAEs) of a related model with parallel boundary fields. These results ... Authors: According to your suggestion, we modified the sentence lines 5-8 in Abstract "Based on the reduced Bethe ansatz equations (BAEs), we obtain the boundary energy of the system. These results ..." into "Based on our findings, the boundary energy of the system in the thermodynamic limit can be obtained from Bethe ansatz equations of a related model with parallel boundary fields. These results ...". 2. Reviewer: Introduction, line 67: delete "unparallel" Authors: Thank you for your kind advice. We deleted the word "unparallel". 3. Reviewer: Section 3, lines 134-137: change to ... is not the eigenvalue $\Lambda(u)$ for any finite $N$ but rather that of the transfer matrix with parallel boundary fields of the same strength. In the limit $N\rightarrow\infty$ it will give, however, the correct boundary energy ... Authors: According to your advice, we changed the sentence "the $\Lambda_{hom}(u)$ is not the eigenvalue $\Lambda(u)$ for any finite... of the paper)." in lines 138-141 of P. 6 to "the $ \Lambda_{hom}(u) $ is not the eigenvalue $\Lambda(u)$ for any finite $N$ but rather that of the transfer matrix with parallel boundary fields of the same strength. In the limit $N\rightarrow\infty$ it will give, however, the correct boundary energy (see the following parts of the paper)." 4. Reviewer: Section 4, line 216/7: delete "unparallel" and add "in the thermodynamic limit" at the end of the first sentence. Authors: Thank you for your kind advice. We deleted the word "unparallel" in line 228 and added "in the thermodynamic limit" at the end of the first sentence in lines 228-229 of P.10.

---

## Round 2 · Referee Report · Anonymous (Referee 3) · 2021-9-25

Strengths

Some new interesting results.

Weaknesses

The main analytic results are derived for boundary conditions different from those declared.

Report

Dear Editor,
The revisited version of the manuscript contains some minor improvements. In particular, some clarification on the numerical (DMRG) results derived by the authors for the case of unparallel boundary conditions, see figure 4. The authors also give some explanation of what the reduced BE are, see their point 8 in the list of changes. Nevertheless, as also remarked by the second referee, they have done very little about the misleading statements and structure of the manuscript. A part the exceptions above remarked, they have mainly leave unattended my previous remarks so that my previous report is largely still actual for the current revision.
The three referees seem all to agree on the fact that the main results of the current manuscript are: a) the analytic computations under the string hypothesis of the boundary energy of the spin 1 open chain under PARALLEL boundary conditions. b) the numerical (DMRG) analysis up to large chains of this quantity for the UNPARALLEL boundary conditions. Then, the comparison of the “analytical/numerical” results in the large chain limit allows to reasonably argue that bulk and boundary energy of the spin 1 open chain should be independent from the specific form of the integrable boundary conditions.
These are interesting and sounding results and this should be what I would expect to understand reading the abstract and all along the manuscript but this is not the case. Instead, to arrive to this conclusion, I have done a careful disentanglement between manuscript statements and analysis there presented and I have the impression that this holds true for the other referees, too. It is enough to look to the abstract:
“The finite size scaling properties of the inhomogeneous term in the T-Q relation at the ground state are analyzed.” This is done only numerically, e.g. by DMRG.
“Based on the reduced Bethe ansatz equations (BAEs), we obtain the boundary energy of the system.” This is done analytically by string hypothesis only for the PARALLEL case. Then, the numerical analysis allows to argue that the same result should hold in the thermodynamic limit also for the UNPARALLEL case.
Let me also cite directly, the opening sentence of section 4:
“In this section, we study the physical effects induced by the unparallel boundary magnetic fields and compute the boundary energy”
Once again, they numerically (DMRG) study the physical effects induced by the UNPARALLEL boundary magnetic fields while they compute analytically by string hypothesis the boundary energy of the PARALLEL case.
As well as the sentence in the conclusion:
“The method provided in this paper can be used to study the thermodynamic properties of other quantum integrable models associated with rational R-matrix.”
It is not clear to me what method? As told, they make analytical analysis of the PARALLEL case, associated to ordinary Bethe equations, while they do numerical (DMRG) analysis for the UNPARALLEL case. Does the statement mean that they expect to be able to make this mixed “analytical/numerical” analysis also for other models and that they expect that also for these models the boundary energy can be argued to be independent from the integrable boundary conditions?
This only to make the point about misleading statements. Let me also recall the problems I cited about the Hamiltonian which are still there as well as an autoreferential attitude about citations.
On this basis I cannot suggest the publication of the manuscript in its current form.
  • validity: ok
  • significance: ok
  • originality: ok
  • clarity: low
  • formatting: reasonable
  • grammar: reasonable

Author:  Xiaotian Xu  on 2021-10-14  [id 1851]

(in reply to Report 3 on 2021-09-25)

Thank you very much for your many helpful suggestions raised about our paper and we really appreciate your support. We believe that these comments and suggestions have significantly improved our manuscript. To these comments we respond as follows. The page numbers and equation numbers refer to revised version, unless specify.

1) Reviewer: Nevertheless, as also remarked by the second referee, they have done very little about the misleading statements and structure of the manuscript. A part the exceptions above remarked, they have mainly leave unattended my previous remarks so that my previous report is largely still actual for the current revision.

Authors: Thank you very much. According to your kind suggestion, we have carefully modified the abstract and corrected the misleading statements in the new revision. For example, we have revised the statements of lines 3-8 in the abstract. We have revised the statements of lines 138-141 in section 3 of P.6. We added some discussions in lines 218-226 of P.10 at the end of section 3. We revised the statements and added discussion of lines 228-232 in section 4 of P.10.

2) Reviewer: These are interesting and sounding results and this should be what I would expect to understand reading the abstract and all along the manuscript but this is not the case. Instead, to arrive to this conclusion, I have done a careful disentanglement between manuscript statements and analysis there presented and I have the impression that this holds true for the other referees, too. It is enough to look to the abstract:

"The finite size scaling properties of the inhomogeneous term in the T-Q relation at the ground state are analyzed." This is done only numerically, e.g. by DMRG.

"Based on the reduced Bethe ansatz equations (BAEs), we obtain the boundary energy of the system." This is done analytically by string hypothesis only for the PARALLEL case. Then, the numerical analysis allows to argue that the same result should hold in the thermodynamic limit also for the UNPARALLEL case.

Authors: Thank you for your recognition of our work and helpful suggestions! We have modified the abstract.

We have changed the sentence in abstract lines 3-5 "The finite size scaling properties of the inhomogeneous term in the $T-Q$ relation at the ground state are analyzed" to "The finite size scaling properties of the inhomogeneous term in the $ T-Q $ relation at the ground state are calculated by the density matrix renormalization group".

We have modify the sentence in abstract lines 5-8 "Based on the reduced Bethe ansatz equations (BAEs), we obtain the boundary energy of the system" to "Based on our findings, the boundary energy of the system in the thermodynamic limit can be obtained from Bethe ansatz equations of a related model with parallel boundary fields".

3) Reviewer: Let me also cite directly, the opening sentence of section 4: "In this section, we study the physical effects induced by the unparallel boundary magnetic fields and compute the boundary energy". Once again, they numerically (DMRG) study the physical effects induced by the UNPARALLEL boundary magnetic fields while they compute analytically by string hypothesis the boundary energy of the PARALLEL case.

Authors: In this paper, we study the physical effects induced by the boundary magnetic fields and compute the boundary energy in the thermodynamic limit, where the system size $N \to \infty$. From the finite size scaling analysis by using the density matrix renormalization group (DMRG) method, we find that the inhomogeneous term indeed can be neglected at the ground state when the system size $N$ tends to infinity. Then we can analytically calculate the boundary energy in the thermodynamic limit based on the string hypothesis of the reduced BAEs (29), which is equal to the result induced by the parallel boundary magnetic fields in the thermodynamic limit. In figure 3, the numerical extrapolation results ($N \to \infty$) of DMRG data also prove the correctness of the analytical results.

We add the related discussion of lines 229-232 of P.10 after the first sentence in section 4.

4) Reviewer: As well as the sentence in the conclusion: "The method provided in this paper can be used to study the thermodynamic properties of other quantum integrable models associated with rational R-matrix." It is not clear to me what method? As told, they make analytical analysis of the PARALLEL case, associated to ordinary Bethe equations, while they do numerical (DMRG) analysis for the UNPARALLEL case. Does the statement mean that they expect to be able to make this mixed “analytical/numerical” analysis also for other models and that they expect that also for these models the boundary energy can be argued to be independent from the integrable boundary conditions?

Authors: Thank you for your helpful suggestion! We modified the related statements in the conclusion as follows.

In this paper, we have studied the thermodynamic limit and boundary energy of the isotropic spin-1 Heisenberg chain with generic integrable non-diagonal boundary reflections. It is shown that the contribution of the inhomogeneous term in the associated $ T-Q $ relation (18) (due to the unparallel boundary fields) at the ground state can be neglected when the system size $N$ tend to infinity. Then we calculate the analytical expression of boundary energy (68) in the thermodynamic limit based on the string hypothesis of the reduced BAEs (29).

5) Reviewer: Let me also recall the problems I cited about the Hamiltonian which are still there as well as an autoreferential attitude about citations.

Authors: Thank you very much. According to your suggestion, we have rewritten the Hamiltonian (15) in a more symmetric form, and we have corrected the problems in the citations. We deleted some self-citations and added five new references:

[39] L. Mezincescu, R. I. Nepomechie and V. Rittenberg, Bethe ansatz solution of the Fateev-Zamolodchikov quantum spin chain with boundary terms, Phys. Lett. A 147, 70 (1990).

[40] T. Inami, S. Odake and Y.-Z. Zhang, Reflection K-matrices of the 19-vertex model and XXZ spin-1 chain with general boundary terms, Nucl. Phys. B 470, 419 (1996).

[45] H.J. de Vega and F. Woynarovich, Method for calculating finite size corrections in Bethe ansatz systems: Heisenberg chain and six-vertex model, Nucl. Phys. B 251, 439 (1985).

[46] A. Kl\"umper, M. T. Batchelor and P. A. Pearce, Central charges of the 6- and 19-vertex models with twisted boundary conditions, J. Phys. A: Math. Gen. 24, 3111 (1991).

[47] J. Suzuki, Spinons in magnetic chains of arbitrary spins at finite temperatures, J. Phys. A: Math. Gen. 32, 2341 (1999).

Author:  Xiaotian Xu  on 2021-10-14  [id 1850]

(in reply to Report 3 on 2021-09-25)

Thank you very much for your many helpful suggestions raised about our paper and we really appreciate your support. We believe that these comments and suggestions have significantly improved our manuscript. To these comments we respond as follows. The page numbers and equation numbers refer to revised version, unless specify. 1. Reviewer: Nevertheless, as also remarked by the second referee, they have done very little about the misleading statements and structure of the manuscript. A part the exceptions above remarked, they have mainly leave unattended my previous remarks so that my previous report is largely still actual for the current revision. Authors: Thank you very much. According to your kind suggestion, we have carefully modified the abstract and corrected the misleading statements in the new revision. For example, we have revised the statements of lines 3-8 in the abstract. We have revised the statements of lines 138-141 in section 3 of P.6. We added some discussions in lines 218-226 of P.10 at the end of section 3. We revised the statements and added discussion of lines 228-232 in section 4 of P.10. 2. Reviewer: These are interesting and sounding results and this should be what I would expect to understand reading the abstract and all along the manuscript but this is not the case. Instead, to arrive to this conclusion, I have done a careful disentanglement between manuscript statements and analysis there presented and I have the impression that this holds true for the other referees, too. It is enough to look to the abstract: "The finite size scaling properties of the inhomogeneous term in the T-Q relation at the ground state are analyzed." This is done only numerically, e.g. by DMRG. "Based on the reduced Bethe ansatz equations (BAEs), we obtain the boundary energy of the system." This is done analytically by string hypothesis only for the PARALLEL case. Then, the numerical analysis allows to argue that the same result should hold in the thermodynamic limit also for the UNPARALLEL case. Authors: Thank you for your recognition of our work and helpful suggestions! We have modified the abstract. We have changed the sentence in abstract lines 3-5 "The finite size scaling properties of the inhomogeneous term in the $T-Q$ relation at the ground state are analyzed" to "The finite size scaling properties of the inhomogeneous term in the $ T-Q $ relation at the ground state are calculated by the density matrix renormalization group". We have modify the sentence in abstract lines 5-8 "Based on the reduced Bethe ansatz equations (BAEs), we obtain the boundary energy of the system" to "Based on our findings, the boundary energy of the system in the thermodynamic limit can be obtained from Bethe ansatz equations of a related model with parallel boundary fields". 3. Reviewer: Let me also cite directly, the opening sentence of section 4: "In this section, we study the physical effects induced by the unparallel boundary magnetic fields and compute the boundary energy". Once again, they numerically (DMRG) study the physical effects induced by the UNPARALLEL boundary magnetic fields while they compute analytically by string hypothesis the boundary energy of the PARALLEL case. Authors: In this paper, we study the physical effects induced by the boundary magnetic fields and compute the boundary energy in the thermodynamic limit, where the system size $N \to \infty$. From the finite size scaling analysis by using the density matrix renormalization group (DMRG) method, we find that the inhomogeneous term indeed can be neglected at the ground state when the system size $N$ tends to infinity. Then we can analytically calculate the boundary energy in the thermodynamic limit based on the string hypothesis of the reduced BAEs (29), which is equal to the result induced by the parallel boundary magnetic fields in the thermodynamic limit. In figure 3, the numerical extrapolation results ($N \to \infty$) of DMRG data also prove the correctness of the analytical results. We add the related discussion of lines 229-232 of P.10 after the first sentence in section 4. 4. Reviewer: As well as the sentence in the conclusion: "The method provided in this paper can be used to study the thermodynamic properties of other quantum integrable models associated with rational R-matrix." It is not clear to me what method? As told, they make analytical analysis of the PARALLEL case, associated to ordinary Bethe equations, while they do numerical (DMRG) analysis for the UNPARALLEL case. Does the statement mean that they expect to be able to make this mixed “analytical/numerical” analysis also for other models and that they expect that also for these models the boundary energy can be argued to be independent from the integrable boundary conditions? Authors: Thank you for your helpful suggestion! We modified the related statements in the conclusion as follows. In this paper, we have studied the thermodynamic limit and boundary energy of the isotropic spin-1 Heisenberg chain with generic integrable non-diagonal boundary reflections. It is shown that the contribution of the inhomogeneous term in the associated $ T-Q $ relation (18) (due to the unparallel boundary fields) at the ground state can be neglected when the system size $N$ tend to infinity. Then we calculate the analytical expression of boundary energy (68) in the thermodynamic limit based on the string hypothesis of the reduced BAEs (29). 5. Reviewer: Let me also recall the problems I cited about the Hamiltonian which are still there as well as an autoreferential attitude about citations. Authors: Thank you very much. According to your suggestion, we have rewritten the Hamiltonian (15) in a more symmetric form, and we have corrected the problems in the citations. We deleted some self-citations and added five new references: [39] L. Mezincescu, R. I. Nepomechie and V. Rittenberg, Bethe ansatz solution of the Fateev-Zamolodchikov quantum spin chain with boundary terms, Phys. Lett. A 147, 70 (1990). [40] T. Inami, S. Odake and Y.-Z. Zhang, Reflection K-matrices of the 19-vertex model and XXZ spin-1 chain with general boundary terms, Nucl. Phys. B 470, 419 (1996). [45] H.J. de Vega and F. Woynarovich, Method for calculating finite size corrections in Bethe ansatz systems: Heisenberg chain and six-vertex model, Nucl. Phys. B 251, 439 (1985). [46] A. Kl\"umper, M. T. Batchelor and P. A. Pearce, Central charges of the 6- and 19-vertex models with twisted boundary conditions, J. Phys. A: Math. Gen. 24, 3111 (1991). [47] J. Suzuki, Spinons in magnetic chains of arbitrary spins at finite temperatures, J. Phys. A: Math. Gen. 32, 2341 (1999).

---

## Round 2 · Author Response

Dear Editor,
Thank you very much for your help. We have revised the manuscript (Ref. No. scipost_202106_00001v1) according to the referees' suggestions. Now we are resubmitting our paper. We think that this paper now meets the requirement of SciPost Physics.
Yours Sincerely,
Xiaotian Xu

---

## Round 2 · List of Changes

We have revised the manuscript according to the referees' suggestions, and list the revisions as follows. The page numbers and equation numbers refer to revised version, unless specify. 1. The word "survives" is replaced with "exists" after Eq.(1) in line 28 of P. 2. 2. In line 33 of P. 2, the sentence "the Hamiltonian (1) degenerates into a projector operator" is replaced with "the Hamiltonian (1) degenerates into a projector operator that is in fact the projection onto the sum of the spin-0 and spin-1 subspaces". 3. The word "Antisymmetry" is changed into "Fusion condition" in Eq.(6) in P. 3. 4. We give the most general off-diagonal boundary matrix $K^-(u)$ in (7)-(8) in P. 3 and P.4, which involves three boundary parameters $p_-,\,\alpha_-,\,\phi_-$. The most general off-diagonal boundary matrix $K^+(u)$ is also given in (10) of P. 4, which has three boundary parameters $p_+,\,\alpha_+,\,\phi_+$. 5. The word "generalized" is replaced with "generated" in line 110 in P. 4. 6. We have corrected the misprint of the definition of the model Hamiltonian (15) in P. 5 and give its expression corresponding to the most general off-diagonal boundary $K$-matrices (7)-(8) and (10). 7. The words "hierarchy fusion" are replaced with "fusion hierarchy" in line 115 in P. 5. 8. We have added some sentences "Throughout this paper...for the i-th and j-th ones." (lines 82-87 in P. 3) to introduce our convention," Some remarks are in order....for any finite $N$." (lines 122-128 in P. 6) to give some remarks, and " It should be emphasized that ....(see the follwing parts of the paper)." (lines 135-138 in P. 6 and P. 7). 9. We have added a footnote in P. 6. 10. The word "subfigure" is replaced with "inset" in line 193 and above line 195 in P. 9. 11. The sentence "Meanwhile, two holes $ \lambda_1^h $ and $ \lambda_2^h $ should be introduced" is replaced with "Note that two holes $\lambda_1^h$ and $\lambda_2^h$ are introduced" in line 230 in P. 11. 12. The words "spin long $z$-direction" are replaced with "spin along the $z$-direction" in line 248, line 253 and line 265 in P. 13. 13. The "$E_b$" is replaced with "$e_b$" in Figure 3 in P. 14. 14. We have changed "...(68) pand the red..." into "...(68) and the red..." in the 2-nd line of the caption of Figure 3 in P. 14. 15. We have added the Figure 4 and the DMRG analysis in lines 271-287 in P. 14. 16. We have rewritten the Conclusions in Section 5 in P. 15. 17. We have deleted some references and added three new ones [33]-[35]. Besides, we have corrected some typos, and words and sentences have also been slightly improved.

---

## Round 3 · Referee Report · Anonymous · 2021-10-30

Report

The authors have implemented the changes requested by the referees. In the present version the abstract and the statements in the main text are in line with the actual results of the paper so that the general acceptance criteria of SciPost Phys are met.

The calculation of the surface energy's dependence on the boundary fields is new. Potentially these results allow for the identification of the boundary conformal field theory describing the continuum limit of the open boundary spin-1 chain (at least for the model with parallel boundary fields).

I recommend to accept the paper for publication in SciPost Phys in its present form.

---

## Round 3 · Referee Report · Anonymous · 2021-11-6

Strengths

Some new and interesting results on spin 1 XXX chain by numeric (DMRG) for the non-parallel case and by analytic analysis of the Bethe Ansatz equations for the parallel case.

Weaknesses

Still some text to improve in particular about references

Report

Dear Editor,
The authors have finally taken into account some of the main remarks presented by the other referees and myself. The manuscript then states more frankly what are the results there derived and the way they are derived, i.e. numerically for the non-parallel case and by analytic analysis of the Bethe Ansatz equations for the parallel case. So, the manuscript can meet the minimal requirements for its publication in SciPost and I will propose its publication once the authors will implement the following further improvements.
About the text:
The Author, should make their modifications compatible with the remaining text. Here, the main point is about the terminology “reduced BAEs”. In line 66 they introduce this terminology without any explanation. I suggest to define there what reduced Bethe equations are, i.e. the Bethe equations of an associated parallel XXX spin 1 model.
About citations:
While some pertinent references are finally included, I think that the manuscript is still deficient of important ones.
i) First of all, one should recall that the so-called TQ-equations, which are heavily used in the manuscript, have been originated by the work of Baxter, reference to the original papers must be added, e.g.
“R. J. BAXTER, Stud. Appl. Math. (Mass. Inst. of Technology) 50 (1971), 51-69.”
“Baxter,R.J.: Partition function of the eight-vertex lattice model. Ann.Phys.70 193-228 (1972)”
ii) The Ansatz functional analysis of the eigenvalue spectrum by TQ-equations has been pioneered and systematically applied to a large class of integrable models by Reshetikhin and it goes under the name of “Analytic Bethe Ansatz”, e.g.:
“Reshetikhin,N.Yu.: The functional equation method in the theory of exactly soluble quantum systems. Sov.Phys.JETP 57 691-696 (1983)”
The analysis developed in the cited papers [28-33] is a natural development of the Reshetikhin’s approach (based on the fusion equations to introduce an Ansatz) once applied to the models under consideration, the same is definitively the case for the so-called ODBA method.
iii) The authors have missed to cite another Ansatz approach, the so-called “Modified Algebraic Bethe Ansatz”:
“S. Belliard and N. Crampé, Heisenberg XXX model with general boundaries: Eigenvectors
from algebraic Bethe ansatz, SIGMA 9, 072 (2013),”
“S. Belliard, Modified algebraic Bethe ansatz for XXZ chain on the segment - I - Triangular
cases, Nucl. Phys. B 892, 1 (2015),”
which allows to overcome the absence of reference states and to have access to an algebraic Bethe Ansatz form of the transfer matrix eigenstates.
iv) Discussing about the analysis of quantum integrable models without U(1) symmetry and without a proper reference states, one is obliged to refer to the seminal works of Sklyanin and its non-Ansatz method of quantum separation of variables:
“E. K. Sklyanin, The quantum Toda chain, In N. Sanchez, ed., Non-Linear Equations in Classical
and Quantum Field Theory, pp. 196–233. Springer Berlin Heidelberg, Berlin, Heidelberg,
ISBN 978-3-540-39352-8”
“E. K. Sklyanin, Functional Bethe Ansatz, In B. Kupershmidt, ed., Integrable and Superintegrable
Systems, pp. 8–33. World Scientific, Singapore, doi:10.1142/9789812797179_0002
(1990)”
Already in the 1985, Sklyanin has derived an algebraic approach to define the spectrum (eigenvalues and wave-functions) of models like the XXX spin ½ chain with general quasi-periodic boundary conditions and the Toda model. Both models for which a reference state is missing and ordinary algebraic Bethe Ansatz does not apply. This SoV approach has since been largely developed in the literature with further recent developments. This is, in particular, the case for the XXX/XXZ/XYZ spin chains with non-periodic boundary condition, see e.g.:
“H. Frahm, A. Seel and T. Wirth, Separation of variables in the open XXX chain, Nucl. Phys.
B 802, 351 (2008)”
“H. Frahm, J. H. Grelik, A. Seel and T. Wirth, Functional Bethe ansatz methods for the open XXX chain, J. Phys A: Math. Theor. 44, 015001 (2011).”

where the functional version (i.e. eigenvalues and wave-functions) of SoV has been derived and see e.g.

“G. Niccoli, Non-diagonal open spin-1/2 XXZ quantum chains by separation of variables:
Complete spectrum and matrix elements of some quasi-local operators, J. Stat. Mech. 2012,
P10025 (2012)”

where the SoV bases have been constructed in the Hilbert space of the quantum spin chain, so allowing for the construction of the eigenstates in the same Hilbert space and providing the tools for a simple proof of the completeness of the spectrum description.

Indeed, one should stress the non-Ansatz nature of the Sklyanin’s SoV approach for which the completeness of the spectrum description is mainly self-contained in it. In fact, it is in the SoV framework that TQ-equations and associated Bethe equations are rederived without any Ansatz and naturally proven to be complete.
This is the case for the inhomogeneous Bethe equations whose Ansatz has been introduced in the ODBA framework:
“J. Cao,W.-L. Yang, K. Shi, and Y.Wang. Off-diagonal Bethe ansatz solutions of the anisotropic
spin-1/2 chains with arbitrary boundary fields. Nucl. Phys. B, 877:152–175, 2013.”
and in reference [32], while a proof of the completeness of the spectrum has been naturally presented in the SoV framework:
“N. Kitanine, J. M. Maillet and G. Niccoli, Open spin chains with generic integrable boundaries:
Baxter equation and Bethe ansatz completeness from separation of variables, J. Stat.
Mech. 2015, P05015 (2014)”
As well as for the simplest example of models without U(1) symmetry, i.e. the XXZ chain with anti-periodic boundary conditions, where the Ansatz on the TQ-equation have been introduced in
“M. T. Batchelor, R. J. Baxter, M. J. O’Rourke, and C. M. Yung, Exact solution and interfacial tension of the six-vertex model with anti-periodic boundary conditions, J. Phys. A: Math. Gen. 28 (1995), 2759.”
“C. M. Yung and M. T. Batchelor, Exact solution for the spin-s XXZ quantum chain with non-diagonal twists, Nucl. Phys. B 446 (1995), 461–484.”
conjecturing the existence of the Q-operator, while proven in the SoV framework in
“G. Niccoli and V. Terras, Antiperiodic XXZ chains with arbitrary spins: Complete eigenstate
construction by functional equations in separation of variables, Lett. Math. Phys. 105, 989
(2015)”

v) Finally in the lines 42-43, where the authors refer to results for periodic and parallel boundary conditions, one should mention the fundamental results on correlation functions for spin 1/2 of the Kyoto and then of the Lyon school. If the authors restrict their discussion to the higher spin case only then it is anyhow worth citing:
“O. A. Castro-Alvaredo and J. M. Maillet, Form factors of integrable Heisenberg (higher) spin chains, J. Phys. A: Math. Theor. 40 (2007), 7451.”
as first results toward the dynamics of the higher spin models.

Requested changes

See report

---

## Round 3 · Referee Report · Anonymous · 2021-11-11

Strengths

1) the vanishing of the contribution of the inhomogeneous term to the ground-state energy is shown

2) new results for the surface energies are derived.

Weaknesses

1) the finite size correction term const.N^beta is not evaluated (analytically)

Report

The authors have taken into account my earlier suggestions. I therefore
recommend publication of the manuscript if the following correction is applied:

In the newly added paragraph on page 10 the authors write

"The finite size correction O(N^1) for the bulk and O(N^0) term for the
boundaries to the ground state energy do not depend on the orientations of the
boundary fields."

The terminology "finite size correction O(N^1)" is inappropriate. Please write

"The O(N^1) bulk term and the O(N^0) boundary term for the ground
state energy do not depend on the orientations of the boundary fields."

Requested changes

see above

---

## Round 3 · Author Response

Dear Editor,

Thank you very much for your help. We have revised the manuscript (Ref. No. scipost_202106_00001v2) according to the referees' suggestions. Now we are resubmitting our paper. We think that this paper now meets the requirement of SciPost Physics.

Yours Sincerely,

Xiaotian Xu

---

## Round 3 · List of Changes

We have revised the manuscript according to the referees' suggestions, and list the revisions as follows. The page numbers and equation numbers refer to revised version, unless specify.
1. We have modified the sentence lines 3-5 in Abstract "The finite size scaling ... are analyzed" into "The finite size scaling properties of the inhomogeneous term in the $T-Q$ relation at the ground state are calculated by the density matrix renormalization group".
2. We have modified the sentence lines 5-8 in Abstract "Based on the reduced Bethe ansatz equations (BAEs), we obtain the boundary energy of the system." into "Based on our findings, the boundary energy of the system in the thermodynamic limit can be obtained from Bethe ansatz equations of a related model with parallel boundary fields.".
3. We have deleted the word "unparallel" in line 69 in P.3.
4. We have rewritten Eq.(15) in P.5 into a more symmetric form.
5. We have changed the sentence "the $\Lambda_{hom}(u)$ is not the eigenvalue $\Lambda(u)$ for any finite ... of the paper)." in lines 138-141 in P.6 to "the $ \Lambda_{hom}(u) $ is not the eigenvalue $\Lambda(u)$ for any finite $N$ but rather that of the transfer matrix with parallel boundary fields of the same strength. In the limit $N\rightarrow\infty$ it will give, however, the correct boundary energy (see the following parts of the paper).".
6. We have added some discussions in lines 218-226 in P.10 at the end of section 3 and added five references [39], [40], [45], [46] and [47].
7. We have deleted the word "unparallel" in line 228 in P.10 and added "in the thermodynamic limit" at the end of the first sentence in lines 228-229 in P.10.
8. We have added some discussions in lines 229-232 in P.10 after the first sentence in Section 4.
9. We have changed the exponent "b" to "$\beta$" in caption of Figure 4 and lines 297, 298 in P.15.
10. We have rewritten the Conclusions in Section 5 in P. 15.
11. We have polished the English.

---

## Round 4 · Author Response

Dear editor,

We are extremely pleased that now all three referees are so strongly supportive of our manuscript (Ref. No. scipost_202106_00001v3). Please pass our great thanks to the referees for their appreciations of our work and also for valuable suggestions which help us to further improve our presentation.

We have given some minor revisions for our manuscript according to the referees' suggestions as follows:

1) We have added the sentences lines 66-79 in P. 2 and 3 to explain the terminology "reduced BAEs".

2) We have added new references [25], [29], [36]-[49] and [51]-[53] (which includes all the references pointed by one of the referees), and given their citations in corresponding positions of the paper.

3) In lines 233-234 of P.10, we have rewritten the sentences as "The O(N^1) bulk term and the O(N^0) boundary term for the ground state energy do not depend on the orientations of the boundary fields".

4) We also give some corrections for the typos.

We are resubmitting our paper for publication in SciPost Physics.

Many thanks for your kind helps and with the best regards!

Sincerely Yours,

Xiaotian Xu

---

## Editorial Decision

publication_decision_taken:_accept